# Molecular Tracking and Remote Sensing to Evaluate New Chemical Treatments Against the Maize Late Wilt Disease Causal Agent, *Magnaporthiopsis maydis*

**DOI:** 10.3390/jof6020054

**Published:** 2020-04-27

**Authors:** Ofir Degani, Shlomit Dor, Assaf Chen, Valerie Orlov-Levin, Avital Stolov-Yosef, Danielle Regev, Onn Rabinovitz

**Affiliations:** 1MIGAL–Galilee Research Institute, Tarshish 2, Kiryat Shmona 11016, Israel; dorshlomit@gmail.com (S.D.); assafc@migal.org.il (A.C.); orlov.valerie@gmail.com (V.O.-L.); avital.stolov@gmail.com (A.S.-Y.); linkar45@gmail.com (D.R.); onnrab@gmail.com (O.R.); 2Faculty of Sciences, Tel-Hai College, Upper Galilee, Tel-Hai 12210, Israel

**Keywords:** Azoxystrobin, *Cephalosporium maydis*, crop protection, field assay, fungus, fungicides, *Harpophora maydis*, pesticides, Real-Time PCR, remote sensing

## Abstract

Late wilt is a destructive disease of corn: outbreaks occur at the advanced growth stage and lead to severe dehydration of susceptible hybrids. The disease’s causal agent is the fungus *Magnaporthiopsis maydis*, whose spread relies on infested soils, seeds, and several alternative hosts. The current study aimed at advancing our understanding of the nature of this plant disease and revealing new ways to monitor and control it. Two field experiments were conducted in a heavily infested area in northern Israel seeded with highly sensitive corn hybrid. The first experiment aimed at inspecting the Azoxystrobin (AS) fungicide applied by spraying during and after the land tillage. Unexpectedly, the disease symptoms in this field were minor and yields were high. Nevertheless, up to 100% presence of the pathogen within the plant’s tissues was measured using the quantitative real-time PCR method. The highest AS concentration tested was the most effective treatment, and resulted in a 6% increase in cob yield and a 4% increase in A-class yield. In the second experiment conducted in the following summer of the same year in a nearby field, the disease outbreak was dramatically higher, with about 350 times higher levels of the pathogen DNA in the untreated plots’ plants. In this field, fungicide mixtures were applied using a dripline assigned for two coupling rows. The most successful treatment was AS and the Difenoconazole mixture, in which the number of infected plants decreased by 79%, and a 116% increase in crop yield was observed, along with a 41% increase in crop quality. Evaluation of the effectiveness of the treatments on the plants’ health using a remote, thermal infra-red sensitive camera supported the results and proved to be an essential research tool.

## 1. Introduction

Late wilt is a disease in corn (*Zea mays*, maize) caused by the fungus *Magnaporthiopsis maydis*, which spreads in the soil and seeds. The pathogen penetrates the plant and blocks the water supply to its upperparts, which can lead to severe dehydration and death of the host plant at the end of the growth session. The disease was first identified in Egypt in the 1960s [1] and is present today in all corn-growing areas of this country [2]. The disease was recognized as the most severe corn disease in this region [3,4]. In Israel, it has existed for about 40 years in the Upper Galilee in the northern part of the country, especially in the Hula Valley, and in the past decade, it has spread to the south. In susceptible maize cultivars and heavily contaminated fields, the pathogen can cause 100% infection and total yield loss [3].

The disease is typified by rapid wilt of the corn plant, which usually takes place two to three weeks before harvesting. Initial infection occurs through the roots, and root discoloration and necrosis can be observed [5,6]. The above-ground symptoms first appear at the tasseling stage [7], and include rapid wilting of the near-ground leaves that progresses upwards during the subsequent two weeks. Yellowish to reddish-brown streaks may appear on the lower internode. Dehydrated stalks take on a shrunken appearance, and the vascular bundles become dark yellow to brownish [8]. Stalk symptoms may be worsened by secondary invaders such as *Fusarium verticillioides* causing stalk rot, and *Macrophomina phaseolina* causing charcoal rot [9], although an antagonistic relationship may exist [6,10]. If kernels are produced, they are often damaged and undeveloped. Eventually, these destructive processes can lead to the plant’s death.

The causal organism of late wilt is the fungus *M. maydis* [11] with synonyms *Cephalosporium maydis* (Samra, Sabet and Hingorani) and *Harpophora maydis* [12]. The genus *Magnaporthiopsis* was established by Luo and Zhang [13] to group three pathogenic species, namely *M. poae*, *M. incrustans,* and *M. rhizophila*, which had previously belonged to the genera *Magnaporthe* and *Gaeumannomyces. M. maydis* was recently transferred to this genus from *Harpophora* by Klaubauf et al. [14]. *M. maydis* reproduces asexually, and no perfect stage has been identified [15]. In the first three weeks of growth, the fungus penetrates the roots, colonizes the xylem tissue, and then grows or is translocated to the above-ground parts of the plants. If the plants are not infected, their susceptibility to late wilt declines from about 50 days after sowing (DAS; [7]).

By flowering (R1-silking, silks visable outside the husks), approximately 50–60 days from sowing, the first symptoms were gradually revealed. Later, *M. maydis* typically colonizes the entire stalk, and the vascular tissue is blocked with hyphae and gumlike deposits, resulting in water supply suffocation and wilting. Indeed, the leaves of diseased plants have a high content of the amino acid proline, probably associated with water stress due to restricted water flow caused by plugging of the tracheary elements [16]. Infection also resulted in a reduction in the number of vascular bundles in a cross-section of the internode. In severe infection, ears can be colonized 12–13 weeks after planting, and the pathogen can colonize the pedicels and move to the pericarp, endosperm and embryo tissues of the seeds [17]. The process of seed infection also occurs in resistant, apparently non-symptomatic, corn cultivars [18]. Kernel infection can result in seed rot and pre-emergence damping-off [19].

The disease progresses rapidly at temperatures of 21–32 °C. Optimal soil pH for *M. maydis* is 6.5, although the fungus can adjust to a wide range of pH environments [20]. Low water potential is considered one of the most influential factors enhancing late wilt disease progression [16,21,22,23]. *M. maydis* can remain viable in the soils as sclerotia for several years. However, the pathogen is considered a poor competitive saprophyte compared to other microorganisms in the soil [24]. Secondary hosts such as lupine, cotton, watermelon, and green foxtail (*Setaria viridis*) can support the survival of the pathogen in the long term [25,26]. In addition to being seed-borne, the pathogen can spread through the movement of crop residues and soils. Land tillage operations can spread the fungus within the field, and tillage equipment can spread it between areas.

In the past, attempts have been made to control the pathogen using different methods, including agricultural (balanced soil fertility and flood-fallowing) [22,27], biological [23,28,29,30,31], physical (solar heating) [32], allelochemical [5], and chemical options [8,33,34]. Notwithstanding the potential of some of these methods, the only means currently applied in Israel and Egypt to date to restrict the disease is the use of resistant corn varieties. Indeed, many sources of resistance have been identified in both countries that have significantly reduced the impact of late wilt in this Mediterranean area. However, more aggressive pathogen strains that can threaten resistant maize cultivars have been identified in Egypt and Spain [35,36].

Previously, applying Azoxystrobin by spraying was found to be ineffective [34]. On the other hand, injecting this fungicide directly into a dripline assigned for each row at three 15-day intervals inhibited the development of wilt symptoms and recovered cob yield by 100%. However, this method was not practical due to the high cost of the irrigation system.

Recently, for the first time in 40 years since the discovery of late wilt disease in Israel, an economically and efficient applicable solution was approved that could be used on a large scale to protect susceptible corn varieties in commercial fields [37]. This solution combines antifungal mixtures with different mechanisms to prevent resistance development and saves about 40% of irrigation costs. The new method was tested in field trials in 2017, and involved the injection into the irrigation system of the substances Azoxystrobin + Difenoconazole (AS + DC, commercial name ‘Ortiva-Top,’ manufactured by Syngenta, Basel, Switzerland, supplied by Adama Makhteshim, Airport City, Israel) at three 15-day intervals. Economic efficiency was achieved by using one dripline for two coupling rows (a row spacing of 50 cm instead of 96 cm). The short row space enables the efficient concentration of the anti-fungal compound in the soil. A recently developed quantitative real-time PCR (qPCR)-based molecular detection system [3] showed that following the AS + DC treatment, the pathogen DNA levels in the host tissue decreased to near-zero values. In the double-row cultivation, this treatment prevented the development of wilt symptoms by 41% and recovered yield to the standard level in healthy fields (1.6 times more compared to the non-treated control). Also, the yield classified as A-class (cob weight above 250 g) increased from 17% in the control to 75% in this treatment.

The current work follows the 2009–2010 [34] and 2017 [37] studies. It aims at addressing three main topics: (i) improve preventive treatments to cope with the disease; (ii) study *M. maydis* pathogenesis by tracking the pathogen DNA in the plants’ inner-tissues and plants’ outer symptoms; and (iii) remote sensing to monitor the plants’ health and evaluate the effectiveness of the treatments.

For examining improved or alternative ways of restricting late wilt disease outbreaks in commercial fields, the study comprised two field experiments: the first (spring, 2018) designed to apply the antifungal compounds by spraying during tillage and at three 15-day intervals (at the early growth stages), as previously tested with dripline irrigation [34]. The second aimed at improving the former method for applying the fungicides using dripline irrigation in double-row cultivation [37] by applying new fungicide mixtures and injecting them in a sequence based on AS + DC in the first application, and the application of pesticides harboring a different action mechanism in the second and third applications.

The effectiveness of these new approaches on the pathogenesis of *M. maydis* studied in an infested field was carried out by evaluating disease symptoms and dehydration levels, measuring yield production and using the qPCR DNA-sequence-based approach. This work also introduced a remote sensing evaluation of the effectiveness of the treatments using a quadcopter equipped with an RGB and infra-red thermal sensitive cameras, a technique recently demonstrated to ensure precise irrigation management [38]. The potential use of thermal aerial imaging to evaluate field crops’ canopy temperature for pre-detecting wilt diseases evolvement and for evaluating the success of prevention treatments was gradually revealed [35].

## 2. Materials and Methods

### 2.1. Field Experiments for Assessing Fungicide Efficiency in Controlling Late Wilt

The field experiments were supervised and authorized by the Consultation Service (Shaham), Ministry of Agriculture and Rural Development, Beit-Dagan, and by the Israel Northern Research and Development (Northern R&D). The landowners, Kibbutz Neot Mordechai and Kibbutz Amir (Upper Galilee, northern Israel), allowed conducting the study on these sites. The field experiments did not include endangered or protected species.

Two subsequent field experiments for assessing fungicide efficiency in controlling *M. maydis* pathogenesis in susceptible cultivars of sweet corn were conducted one after the other during the spring and summer of 2018. These two subsequent experiments were performed in two nearby fields (located about 10 km from each other) in the Hula Valley in the Upper Galilee, northern Israel, which have both been known to be late wilt infested for many years [34,37]. Both experiments aimed at evaluating Azoxystrobin as a sole fungicide or in combination with Difenoconazole, examining new ways of applying the preparation, assessing its effectiveness in comparison to other pesticides, and developing a method that would prevent the development of resistance against the preparation. Both experiments had a similar experimental design with 10 repeats for each of the treatments applied. As expected, the average temperature and humidity measurements in the summer experiment were slightly higher (Table 1), but overall, the conditions were similar and within the range enabling late wilt disease to develop.

#### 2.1.1. The Spring 2018 Field Experiment for Assessing Azoxystrobin Spraying during and after Land Tillage

The first experiment aimed at inspecting the AS applied by spraying during the land tillage stage before sowing and at three intervals after sowing at different dosages. Kibbutz Naot Mordechai’s (Hula Valley in the Upper Galilee, northern Israel) southern area maize field (plot no. 10) was chosen for the experiment. This field was sown with sweet maize Jubilee cv. (developed by Pop Vriend Seeds B.V., Andijk, The Netherlands, supplied by Eden Seeds, Reut, Israel). The Jubilee cultivar had previously been tested for susceptibility in the same maize field and proved to be late wilt-sensitive [34].

Plots in the field were arranged in a complete randomized block design. The experiment comprised 90 plots, including untreated control and eight different chemical treatments. The number of repeats (plots) involved 10 plots per treatment or control. Each plot contained two rows, and each row was 12 m long and contained seven maize plants m^−1^ (approximately 84 plants per row). Row spacing was 96.5 cm. Seeds pretreated with Thiram, Captan, Carboxin, Metalaxyl-M (manufactured by Rogers/Syngenta Seeds, Boise, ID, USA, were supplied by CTS, Tel Aviv, Israel, quota NC7323XLF). The field was watered twice a week with a frontal irrigation system (600 mm per growth season in total). Seeding was performed on 23 April 2018, and germination (with a frontal irrigation system) one day later. Plants emerged above the ground surface six days after planting. Plants first pollinated when they reached 70% silk on 12 June 2018 (50 DAS). The pollination continued for one week. The field was harvested on 5 July 2018 (73 DAS).

Field plots were treated separately with Azoxystrobin fungicide according to the schedule described in Table 2. Applying the AS fungicide during the land tillage stage was done by spraying, followed by mechanical mixing of the top 10–15 cm of the ground before sowing. The control plots were tilled in the same way, but no fungicide was applied. The chemical spraying during the growth session was done with an electric backpack field sprayer, and treatment was applied to the base of the stem. Immediately after each spraying treatment, the field was watered using a frontal irrigation system to enable the fungicide to absorb through the root system. Each spraying treatment was applied three times at a dosage of 2.5 L/hectare, on May 7 (phenological stage V2, four visible leaves, 25 cm plant height), May 22 (phenological stage V4–5, 8–10 visible leaves, 40 cm plant height) and June 3 (phenological stage V8, 12–14 visible leaves, 110 cm plant height), 13, 29, and 41 DAS.

Plant emergence evaluation was conducted 17 DAS for five randomly selected representative plots within each treatment. The *M. maydis* target DNA qPCR detection was done for five randomly selected representative plants from each treatment and the control, in the roots 29 DAS, and in the lower stem tissue 58 DAS (3 DAF) and 73 DAS (18 DAF). Dehydration was determined on the last sampling day for the qPCR detection (73 DAS). Yield quantity (kg cobs per m^2^) and yield quality (average cobs weight) were determined 19 DAF (74 DAS) and comprised all the upper part plant cobs in each of the experimental plots.

#### 2.1.2. The Summer 2018 Field Experiment for Assessing Seed Coating and Fungicide Application by Dripline Irrigation

The second experiment was performed in the nearby Kibbutz Amir maize field (southern area of the maize field, Mehogi 5 maize plot) in the Hula Valley (Upper Galilee, northern Israel). This subsequent field experiment aimed at assessing AS + DC seed coating and the use of five different commercial fungicides implemented through a drip irrigation line. The sweet maize Prelude cv. was chosen as a representative susceptible cultivar. This cultivar had been previously tested for susceptibility in the same maize field (and was found to be late wilt-sensitive, similar to the Jubilee cv.), and the maize field had been reported to be infested with the late-wilt pathogen for many years [3]. This area was a segregated part of a maize field used for grain production. The cultivations comprised the use of a double-row garden bed (row spacing of 50 cm). In this cultivation method, the plants were irrigated using one dripline for two adjacent rows to ensure a sufficient fungicide supply, whose spreading in the ground from the drip irrigation point to the plant is limited. This method is considered more cost-effective due to the lower cost involved in laying one dripline for two adjacent rows instead of assigning one dripline per row, as was previously done [34].

Each of the five chemical treatments and the unprotected control included 10 independent repeats (plots). The total area of each treatment was 386 m^2^, and the combined experimental area was 0.23 hectares. Plots in the field were arranged using a complete randomized block design. The experiment included 60 plots each consisting of two rows. Each row was 20 m long and comprised eight maize plants m^−1^ (ca 160 plants per row).

The field was irrigated with a 16 mm dripline at a 50 cm drip spacing (Dripnet PC1613 F, Netafim, Israel). The drip flow rate was 1 L/H. Overall, throughout the growing season, the field was watered with a total of 600 mm (100 mm immediately after the seeding, and the other 500 mm was dispersed equally throughout the growth session). All five treatments and the control were irrigated with an automatic irrigation controller (NMC Junior, Netafim, Israel). For insertion and mixing the fungicides with the irrigation water to a homogeneous blend in each of the treatments, a five-outlet manifold (with a specific unit at each outlet) was used.

Seeds pretreatment was done according to a standard commercial procedure by Gadot Agro, Kidron, Israel, with a 0.002 cm^3^/seed AS + DC commercial preparation (Ortiva top, see Table 3), as recommended by the manufacturer – Syngenta AG, Basel, Switzerland. Sowing was performed on 21 June 2018, and germination occurred one day later by watering the field. Plant emergence above the ground surface was recorded approximately six days after seeding. In the dripline protected plots, the Prelude cv. plants were treated individually with different commercial fungicides (Table 3): Azoxystrobin and Difenoconazole mixture (Ortiva top, AS + DC); Azoxystrobin (Amistar, AS); Difenoconazole (Dividend, DC) and a mixture of 26.7% Boscalid and 6.7% Pyraclostrobin (Signum W.G., BC + PS). The fifth treatment was an alternation of fungicides according to the following sequence: (1) AS + DC; (2) Prothioconazole (Proline) and Tebuconazole (Folicur) mixture (PR + TE); and (3) Fluopyram (Velum) and Trifloxystrobin (Flint) mixture (FL + TR).

Each fungicide or mixture was applied three times on 9 July, 22 July and 5 August 2018 (18, 31 and 45 DAS, respectively) at a dosage of 2.25 L/hectare (0.0869 L/treatment). The control treatment included plants protected only by the AS + DC seed coating. Flowering occurred on 10 August 2018 (50 DAS), and pollination was performed when the plants reached 70% silk on 15 August 2018 (55 DAS). Wilting symptoms in the control plants were first revealed approximately at this age (55 DAS), and wilt determination was carried out for all of the experimental plants 71 DAS, 16 DAF (31/8/18) by estimating the percentage of plants’ upper leaves having late wilt symptoms of dehydration. These include leaves changing color to light-silver and then to yellow-brown, and rolling inward from the edges. Yield determination comprised all the upper part plant cobs in a 5-m-long section of each row. The cobs of each row were weighed independently. Until harvest day (5 September 2018, 21 DAF, 76 DAS), plots in the control treatment had collapsed, and severe yield loss recorded. 

### 2.2. Remote Sensing for Evaluating the Efficacy of Treatments Based on High-Resolution Visible-Channel and Thermal Aerial Imaging of the Cornfields

A DJI Phantom 4 advanced quadcopter unmanned aerial vehicle (UAV) was used as the RGB flying platform (DJI, Nanshan, Shenzhen, China). The UAV was equipped with a built-in RGB camera having a 4864 × 3648 pixel 4K resolution CMOS sensor and a 24 mm (35 mm eq.) lens with a field of view (FOV) of 84° in a 3-axis stabilized gimbal (https://www.dji.com/phantom-4-adv/info). DJI Inspire-I UAV was used as the thermal flying system, supplied with a fully stabilized gimbal control thermal camera (Zenmuse-XT radiometric), having a 640 × 512-pixel recording size, 32° FOV lens, and a spectral band of 7.5–13.5 µm with both RJPEG and TIFF file formats (https://www.dji.com/zenmuse-xt/info). The flight program was created with the Pix4Dcapture software (Prilly, Switzerland), which was also used as an automatic pilot of both the DJI Phantom 4 Advanced and the DJI Inspire-I UAVs. The flight path had an overlap percentage of 70% and 90%, respectively, to ease the task of mosaicking. The images collected during each flight campaign were georeferenced and mosaicked using Pix4Dmapper software (Prilly, Switzerland) for the RGB images and Pix4DFields (Prilly, Switzerland) for the thermal images.

The cornfields were scanned at low altitudes to obtain high-resolution images, pixel sizes ranging from 0.03–0.04 m. Spatial high resolution is essential in order to distinguish between vegetation pixels and bare soil pixels, therefore allowing to measure the temperature of vegetation accurately without noise. This thermal aerial mapping technique aims at monitoring the foliage temperature of corn plants by measuring the conductivity of the transport tissue. This method may also report on the development of the disease even before the onset of overt symptoms [35]. RGB imaging was used to create the green-red vegetation index (GRVI) that correlates to plant health status [38]. Spectral vegetation indices built from joining channels at various wavelengths allow for improved information extraction from the data collected by remote sensing. This is because they reduce the effects of view angle, soil and topography while intensifying the focus on vegetation visibility.

In the experiment conducted in the Neot Mordechai field, we examined the change in plant foliage temperature in the experimental groups and in the untreated control plots after 34, 41, 52, 59, 65 and 71 days from sowing at mid-day (12:00–14:00). However, since the disease outburst was very weak, no significant difference in plant temperature between the treatments and the control could be measured (hence the results are not presented). In the summer experiment conducted in Kibbutz Amir, two flight campaigns were made to track changes in plant foliage temperature in the experimental groups and in the untreated control plots. This evaluation was made 63 and 73 days from sowing (8 and 18 DAF, respectively) at mid-day (12:00–14:00).

### 2.3. Molecular Diagnosis of the Late Wilt Pathogen

#### 2.3.1. Plant Material

Five to ten plants were collected arbitrarily from each treatment according to the schedule described above. The plants were collected from scattered places along the row, and special care was taken to ensure they had the same phenological development and disease characteristics as their nearby plants (thus, they well represent the average plants in their surroundings). Plant tissues were washed of visible soil by rinsing thoroughly under running tap water. Tissues were sampled by cutting a cross-section of approximately 20 mm in length from each plant. Total tissue fresh weight was adapted to 0.7 g and considered as one repeat. Plant samples were inserted with 4 mL CTAB buffer into universal extraction bags (Bioreba, Reinach, Switzerland), and the tissue was ground to homogeneity for 5 min using a hand tissue homogenizer (Bioreba, Reinach, Switzerland). Finally, the DNA samples were processed for purification, as described below.

#### 2.3.2. DNA Extraction and qPCR

##### DNA Extraction

DNA was extracted and purified from tissue samples of maize tissue known to be infected with *M. maydis* and from axenically grown maize tissue using a minor modification of the procedure of [39]. After grinding the tissue sample with 4 mL CTAB buffer (0.7 M NaC1, 1% cetyltriammonium bromide [CTAB], 50 mM Tris-HC1 pH 8.8, 10 mM EDTA and 1% 2-mercaptoethanol), 1.2 mL was incubated for 20 min at 65 °C. The samples were then centrifuged at 14,000 rpm at room temperature (24 °C) for 5 min. Immediately afterwards, the upper phase lysate (usually 700 µL) was extracted with an equal volume of chloroform/isoamyl alcohol (24:1). After vortex mixing, the mixture was centrifuged again at 14,000 rpm for 5 min at room temperature. The chloroform/isoamyl alcohol extraction stage was repeated twice. Subsequently, the supernatant (usually 300 µL) was separated into a new Eppendorf tube and mixed with cold isopropanol (2:3). The DNA solution was blended gently by inverting the tube several times, maintained at −20 °C for 20–60 min and then centrifuged (14,000 rpm for 20 min at 4 °C). The precipitated DNA was isolated and resuspended in 0.5 mL 70% ethanol. After additional centrifugation (14,000 rpm at 4 °C for 10 min), the precipitate DNA was separated and dried overnight in a sterile hood. At this stage end, the DNA was suspended in 100 µL HPLC-grade water and maintained at 20 °C until use.

##### qPCR-Based Method

The real-time PCR reactions were executed as described previously [3] using the ABI PRISM 7900 HT Sequence Detection System (Applied Biosystems, Massachusetts, CA, USA) and 384-well plates. Conditions of the qPCR were as follows: 5 µL total reaction volume was used per sample well – 2 µL of DNA sample extract, 2.5 µL of iTaq™ Universal SYBR Green Supermix (Bio-Rad Laboratories Ltd., Rishon Le Zion, Israel), 0.25 µL of forward primer and 0.25 µL of reverse primer (to a well 10 µM from each primer). The qPCR cycle plan was as follows: precycle activation stage, 1 min at 95 °C; 40 cycles of denaturation (15 s at 95 °C), annealing and extension (30 s at 60 °C), and finalizing by melting curve analysis. The plant’s root and stem samples from each experiment were analyzed separately by qPCR. The A200a primers were used for qPCR (sequences detailed in Table 4). The gene encoding the eukaryotic mitochondria respiratory electron transport chain, last enzyme – cytochrome c oxidase (*COX*) – was used as a reference “housekeeping” gene to normalize the amount of DNA [40]. This gene was amplified using the primer set COX F/R (Table 4). Calculating the relative gene abundance was according to the ΔCt model [41]. The same efficacy was assumed for all samples. All amplifications were performed in triplicate.

### 2.4. Statistical Analyses

A fully randomized statistical design was used to analyze the *M. maydis* infection outcome on symptoms in the field plants. Data analysis and statistics were done using the JMP program, 15^th^ edition, SAS Institute Inc., Cary, NC, USA. The one-way ANOVA followed by multiple comparisons posthoc of the Student’s *t*-test for each pair (with correction for multiple comparisons) was used to evaluate the *M. maydis* infection outcome in the experiments and the effectiveness of the treatments. In the field trials’ molecular DNA tracking, a high level of variations exists within the results due to changes in environmental conditions, and the non-uniform spreading nature of the late wilt disease pathogen [43]. Consequently, relatively high standard error values resulted, and in most of those tests, no statistically significant differences could be measured.

## 3. Results

By using molecular targeting of the pathogen DNA inside the plants’ tissues, above-ground symptoms evaluation, and new remote sensing of the plants’ health, we conducted two large field experiments to inspect new chemical treatments against the maize late-wilt pathogen, *M. maydis*. The preventive treatment includes applying the fungicide during and after land tillage by spraying it on the base of the stem at a timetable fitting the pathogen lifecycle key points that succeeded earlier in restricting the disease outburst. This application is important to the growth area in which watering is done using a frontal irrigation system. The current work also tested the application of different anti-fungal formulations through irrigation driplines and the alternation of pesticides in order to prevent the development of fungicide resistance.

### 3.1. The Spring 2018 Field Experiment for Assessing Azoxystrobin Spraying during and after Land Tillage

The addition of AS was done during the land tillage phase before sowing and afterwards in three intervals (13, 29, and 41 DAS), as detailed in Table 2. The treatments varied by increasing the amounts of AS fungicide up to the maximum value of 12.5 L/hectare during land tillage, with the addition of three dosages of 2.5 L/hectare applied later by spraying. Measuring the number of emerging plants per square meter at 17 DAS led to the conclusion that none of the chemical treatments were toxic to the plant regarding this measure (Table 5). The development of the field maize plants and the appearance of disease symptoms were monitored throughout the growing season. However, unexpectedly, the disease did not evolve, and the disease symptoms were few and did not affect the growth parameters (plant size, phenological development, and the plant’s above-ground parts color).

To elaborate on this, all these growth aspects were identical in control groups, and the treatments and plants of the entire experimental field had a healthy appearance and normal development; thus, an estimation of nearly 100% healthy plants in all of the treatments and the control was determined. Demonstrating this, are the results collected at the end of the growth season (73 DAS, 18 DAF), by close inspection (Figure 1) and from aerial scans (Figure 2). Symptoms in the plants of the most intensive treatment (12.5 L/hectare spraying during land tillage and later three dosages of 2.5 L/hectare) were very similar to symptoms in the control group (Figure 1) and symptoms of the other treatments groups. The symptoms included partial and occasional color alternation (from green to yellow or brown) of the first above-ground internode and dehydration of the first one or two leaves. Above this lower stem section, the disease symptoms rarely appeared, and, if present, were minor and had no effect on plant growth parameters. Evaluation of the efficiency of the treatments using a remote sensing quadcopter equipped with a thermal infra-red sensitive camera supported the results of the visual estimation of the disease symptoms. No measurable differences were observed between the control group and the chemical treatments or between the chemical treatments themselves (Figure 2).

The number of plants per m^2^ (emergence) measured 17 DAS show similar values of all treatments, except for significantly high value (*p* < 0.05) of the 10 L/hectare during land tillage, compared to relatively low values in three other treatments (Table 5). The yields achieved in this experimental field, regardless of the treatment applied, were very high (without statistical significance among the treatments). The yield evaluation ranged from 2.32 kg/m^2^ in the treatment of two intervals of 2.5 L/hectare by spraying (13 and 29 DAS, without intervention during land tillage) to 2.73 kg/m^2^ in the most intensive treatment (12.5 L/hectare during land tillage followed by three spraying treatments later of 2.5 L/hectare) (Table 5). In the untreated control plots, cob yield reached a value of 2.58 kg/m^2^. For comparison, the control of a field experiment carried out in 2010 at the same location with the same maize cultivar (Jubilee) was severely affected by the disease, collapsed, and yielded only 1.02 kg/m^2^ of cobs [34]. Even in comparison to the disease-free fields planted with the same maize cultivar, which yielded 1.8 kg/m^2^ [18], the control group in the current 2018 spraying experiment resulted in remarkably higher results (Table 5). A statistically insignificant addition of 6% of cob yield resulted in the highest levels of chemical treatment applied. This improvement was also reflected by a 4% increase (not statistically significant) in yield quality (number of A-class cobs that weighed more than 250 g).

Interestingly, regardless of the absence of symptoms, tracking the *M. maydis* DNA inside the host plant tissues revealed that the pathogen was established and developed inside the plants. Overall, the pathogen DNA levels inside the experimental plants’ stems increased gradually during the session. At 29 DAS in the roots, most treatments managed to reduce the relative amounts of fungal DNA to undetected levels (Table 6). From day 58 onwards (in the stem), almost all the treatments were infected (percentages of infection were 20%–100%). The maximum efficiency (reducing the pathogen relative DNA levels) was achieved in the 5 L/hectare spraying during the land tillage treatment (with no additional chemical intervention along with the growth session), which led to undetectable amounts of the pathogen DNA at 73 DAS. Interestingly, in this treatment at 29 DAS, a significantly (*p* < 0.05) high pathogen DNA levels were recorded. In the most intensive treatment (12.5 L/hectare spraying during land tillage and three dosages later of 2.5 L/hectare), an opposite tendency was measured with 100% (58 DAS) or 60% (73 DAS) of the plants sampled infected and 100 times higher *M. maydis* relative DNA levels compared to the control at the session ending in the stem (Table 6). The untreated control plots had 20% measurable infection (2.1 × 10^−5^ relative *M. maydis* DNA levels) starting from the first measurement in the roots 29 DAS, with an increase to double the number of infected plants (1.5 × 10^−4^ relative *M. maydis* DNA levels) in the stem at 58 DAS and a decrease in the stem at the session end (73 DAS, 20% infected plants, 6.5 × 10^−5^ relative *M. maydis* DNA levels). Overall, the stem relative *M. maydis* DNA levels in the control plots were similar or up to 10 times higher than measurements of the chemical treatments at 58 DAS but were lower than most treatments at 73 DAS (up to 4308 times less in comparison to the most prominent treatment in this measure, 10 L/hectare during land tillage).

### 3.2. The Summer 2018 Field Experiment for Assessing Seed Coating and Fungicide Application by Dripline Irrigation

The effect of an integrated treatment of seed coating and drip irrigation fungicide application against *M. maydis* in the field was assessed in order to improve the current protocol [37]. All the experimental groups had the same above-ground surface appearance during the initial growth stage (Table 7) as expected since no treatment was applied until that growth day. As in the spring 2018 experiment described above, in the fallowed summer experiment, roots and shoot fresh weight measurements taken 30 DAS supported the inference that no apparent phytotoxicity resulted from the first chemical irrigation treatment (applied at 18 DAS). On the contrary, some of the treatments resulted in higher (but not statistically significant) roots and shoot biomass. Also, the flowering and fertilization dates (50 and 55 DAS, respectively) were regular, as anticipated.

Initial signs of the disease appeared around 55 DAS near the male flowering. At the age of 71 DAS (16 DAF), most of the control (protected only by AS + DC seed coating) Prelude cv. plants (72%) were diseased and wilting (Figure 3, Figure 4 and Figure 5). In agreement with the literature [7,18], the stem surface symptoms in diseased plants included color alternation of the lower stem near the first above-ground internode (Figure 4). Indeed, in the control, DC, alternation, and BC + PS treatments (in most plants), the upper first internode part or its surrounding leaf was arid (Figure 4, right panel). In contrast, most of the AS treatment plants showed only slight symptoms, and the AS + DC plants had a healthy appearance in this lower section. The cobs’ symptoms (Figure 4, left panel) supported the lower stem symptoms described above. Wilt symptoms were observed on the leaf, whose cob is located in its axil, in the same treatment that had color alternation (to yellowish-light brown) of the lower internodes. The qualitative evaluation (Figure 3) was supported by the quantitative assessment (Figure 5). The AS embedded treatments all resulted in a significant (*p* < 0.05) reduction in wilting symptoms or elevation in the healthy plant’s percentages, compared to the control. In particular, the AS + DC treatment excelled in rescuing the plant’s health with nearly 100% recovery. The DC treatment had some beneficial influence (not statistically significant) with a 32% improvement in the healthy plant’s percentages, whereas the BS-PC was very similar to the control in this measure.

The thermal measurement from the air using the remote sensing quadcopter equipped with an infra-red sensitive camera supported the ground evaluation. The sensitivity of this method allows detecting temperature variations of less than 1 °C degree. Indeed, using this method, a signal reduction (with no statistical difference from the control) was measured in the canopy temperature of the AS + DC treated plants (Figure 6). Still, the thermal measurement sensitivity needs to improved for evaluating successful treatments against late wilt (Figure 6). A similar reduction was measured in the AS and the fungicide alternation treatments. Both the DC and BS+PC treatments plants had similar canopy temperatures to the non-protected control plants.

To exemplify the thermal measurement, analysis of block 5 (from the 10 blocks or repeats) of this experiment on day 63 (8 DAF), is demonstrated (Figure 7). In this repeat, the AS treatment had the most prominent influence, reducing the canopy temperature by 1.2 °C. This difference can be seen in the RGB air photo (Figure 7B) in which the AS treatment has a greenish appearance of the canopy with no yellow-brown patches (that are present in the more yellowish control treatment). In the thermal photograph, this difference is observed as a dark-blue (colder temperature) appearance of the AS-treated rows while the non-drip-protected control rows have a higher temperature represented by reddish-pink color. Here, too, the DC and BS+PC had canopy temperatures similar to the control.

The yield production (Figure 8) and yield quality (Figure 9) measured 76 DAS (21 DAF), both clearly demonstrated the success of the AS + DC, AS, and anti-fungal compounds alternation treatments, compared to the control plots. These three treatments led to 116%, 92%, and 60% elevation in cob yield, respectively (statistically significant from the DC treatment and the control, p < 0.05). The BS-PC treatment led to a 24% lesser (not statistically significant) improvement in yield quantities, while DC had no influence and resulted in similar yield production to the control (Figure 8). The AS + DC drip irrigation advantage was reflected not only in cob yield but also in cob quality. In this treatment the A-class yield increased to 14% (1.4 times higher compared to the control) in comparison to the control (Figure 9). Moreover, the two other AS containing treatments (AS and the alternation protocol) had higher A-class yield values, compared to the control, and a significant (*p* < 0.05) B-class yield elevation. The BC + PS and the DC treatment results were similar to the control.

At the end of the growth session (71 DAS), all the chemical protective treatments that contained AS (especially AS + DC) reduced the number of infected plants according to the qPCR evaluation. The molecular data are in agreement with the disease symptoms recording (Figure 5) and yield results (Figure 8 and Figure 9). The achievement of the AS + DC treatment in limiting the disease symptoms’ severity (Figure 2, Figure 3, Figure 4 and Figure 5) was reflected in the pathogen DNA spread from day 53 in the stem onwards (Table 8). DNA extracted from the plant samples collected at 53 and 71 days from sowing for subsequent analysis using the qPCR-based technique revealed that the AS + DC treatment reduced the infected plant’s percentage by 33% and 50%, respectively. The fungal DNA levels (reflected as *Mm/cox* ratio) on both dates (53 and 71 DAS, Table 8) in the non-protected control plants (3.5*10^−5^ and 0.02, respectively) were higher than the control group values in the spring spraying experiment at the session end (1.5 × 10^−4^ and 6.5 × 10^−5^ on days 53 and 71, Table 6). At 71 DAS, this elevation resulted in about 308 times higher levels of the pathogen DNA in the untreated control plants’ stems.

## 4. Discussion

Until lately, restricting late wilt disease of maize was very limited, and despite intensive efforts carried out over the past four decades, none of the various potential methods proposed for reducing disease damages were applied in commercial fields in Israel. Recently, a combined treatment of AS + DC seed coating and dripline irrigation (three dosages in 15-day intervals from sowing) managed to prevent disease outbreaks and resulted in crops recovered to the degree expected in healthy fields [37].

The Neot Mordechai field assay (spring 2018) presented here was conducted with some similarities as in another field experiment previously carried out [34] at the same location with the same maize hybrid and the same chemical applied at the same timetable. However, the new experiment evaluated spraying the fungicide instead of applying it through the dripline irrigation system and included pretreating the soil during the land tillage stage. This application is more suitable to fields where watering is based on a frontal irrigation system, as in the Hula Valley (Upper Galilee, northern Israel). Interestingly, in the original field experiment conducted in 2009–2010, the disease outburst was harsh and led to severe dehydration of the untreated plants, whereas in the experiment reported here, the disease symptoms were very weak and had almost no influence on yield productivity, which was very high.

As mentioned in the description of the results, maize crop yield of over 2 kg/m^2^ is considered normal in healthy fields. The spring 2018 experiment (Neot Mordechai field) achieved high yields of 2.5 kg/m^2^, but in the subsequent 2018 summer experiment (Amir field), the untreated controls resulted in poor yields of 0.25 kg/m^2^ (Figure 8), and even the most effective treatment, AS + DC drip protection, recovered cob yield to only 0.54 kg/m^2^. Despite this significant improvement in yield value, this harvest production is still considered relatively low in commercial fields. Since there were no drastic changes in weather conditions (Table 1), this difference between the spring and summer minor diseased plots was probably the result of other factors. These factors are likely the result of the environment since both maize cultivars studied here (the Jubilee cv. and the Prelude cv.) can reach similar yield production (kg/m^2^), as presented in this work and previous work [37]. Indeed in the summer experiment, the field suffered from another fungal disease caused by *Fusarium verticillioides* and *Fusarium oxysporum*, which led to dehydration and yield loss. Nevertheless, in the plants that were not affected by the *Fusarium* spp. disease, the drip protection with AS + DC abolished almost completely any sign of late wilt disease. This effect is probably not reflected in the interpretation of the results since the control group was used as a reference group, and all experimental plots were compared to this control. Thus, if any other factors, such as other pathogens and environmental conditions, are reflected in the planet’s health, development and yield, they are common to the control and all the treatments and equally reflected in the results.

Tracking the *M. maydis* DNA fluctuations using the qPCR method proven earlier [3] exhibited interesting results. By examining only the control treatment in both field experiments conducted here (spring and summer 2018), a different pattern is revealed (Figure 10). In both experiments, the pathogen DNA in the stem at 53–58 DAS was higher than the levels in the roots at 30–31 DAS. However, in the healthy state Neot Mordechai field plants (spring 2018), this trend changed, and *M. maydis* DNA levels dropped at the session end (73 DAS). In contrast, in the heavily diseased Amir field in the summer of the same year, the pathogen’s DNA levels intensified towards the session end (71 DAS). It is logical to assume that the decrease in DNA levels in the non-diseased plants and the sharp increase in these levels in the diseased plants are related to plant health and its immune system ability to prevent fungal spread. Indeed, it was shown that the susceptibility of sensitive maize cv. to late wilt decreases with age [7]. Thus, recovery from late wilt in slightly or moderately infested areas may be associated with a decrease in fungal DNA levels inside the host tissue. In contrast, the weakening of the plants in severe cases of late wilt may lead to the opposite tendency, with sharp fungal DNA levels elevation within the host. To support this, comparing the summer field experiment 2018 results to the summer 2017 results (conducted at the same site) published earlier [37] indicates that in both cases, the harsh disease outbreak was accompanied by a sharp elevation in fungal DNA inside the stem tissue (Figure 10).

A more complex picture regarding the *M. maydis* DNA spreading and late wilt symptoms severity may exist. In a field experiment conducted by our research group in the same fields in the Hula Valley (Upper Galilee, northern Israel) using qPCR, we measured a gradual reduction in detectable *H. maydis* DNA inside the host stem towards 70 DAS at the stage when the plants show extreme dehydration [3]. This DNA reduction infers a decrease in fungal’s cells biomass, which may result in degradation of the fungal DNA. We hypothesized that at the growth session end when the maize host tissues dry out, the fungus enters into the asexual reproduction stage and develops spores and sclerotial bodies [44]. At the same time, the primary hyphae biomass gradually comes apart. Thus, the relationship between symptom development and DNA quantity should be examined more deeply in future studies.

Remote aerial thermal infra-red imaging of crop canopy was tested previously for precision irrigation management [38], but this is the first use of this method in Israel to detect early symptoms of late wilt and to evaluate the effectiveness of preventive treatments. The thermal imaging capable of detecting delicate variations in the measured canopy’s temperatures (usually less than 1 °C degree, Figure 6 and Figure 7) and proved to be reliable and accurate. However, it still needs to be improved to achieve greater sensitivity. Since maize is a summer crop and the main symptom of late wilt is dehydration, this affects water conductivity and results in elevated temperatures of the above-ground plant parts. Measuring these variations at mid-day before scheduled irrigation (when the plant’s respiration is most sensitive to the availability of water) can give a broader overview of the field’s health and highlight regions that are more susceptible to the disease burst, encouraging intervention by agro-mechanical means (for example, excessive watering). The importance of this is significant because the disease is typified by a delayed wilting of affected plants [11] and due to the scattered appearance of late wilt disease in the field [43]. The canopy temperature measure is also a useful means for assessing *M. maydis* isolate aggressiveness, as demonstrated in Spain [35].

It was previously shown that *M. maydis* DNA is present in the host tissues of effectively treated plants [18]. This suggests the potential risk that the fungus will become resistant to fungicides. Thus, incorporating two or more active ingredients with a different mode of operation may be essential in the long term for avoiding the acquirement of fungal immunity. Azoxystrobin is a member of the Qo-inhibiting fungicides (QoIs) class, which is one of the essential classes of agricultural fungicides [45]. However, the growing incidences of resistance to these fungicides and the resulting control failure are becoming a major problem (see, for example, in *Magnaporthe grisea* [46]). Azoxystrobin resistance has been recognized to occur in different fungal species and target site mutations in cytochrome b gene (G143A, F129L) and additional mechanisms. Cross-resistance has also been reported to occur among all members of the QoI group (Fungicide Resistance Action Committee [FRAC] Code List^©^ 2018). Thus, integrating Azoxystrobin with other fungicides is crucial for late wilt control. This work provides such a solution using the integration of pesticide mixtures (with a different mode of operation) and injecting these mixtures through the drip irrigation lines, alone or in alternation in a time table adapted to key points in the disease development. Together with the successful treatment protocol based on changing the cultivation method [37], the method demonstrated here (Amir field, summer 2018) can now be applied in commercial fields for the protection of sensitive corn varieties against late wilt.

## 5. Conclusions

The current work is part of the search for maize late wilt disease management practices, including chemical treatments and cultural management. This effort is of immense importance in containing the spread of disease epidemics and in minimizing economic loss for field crops. This work advances our understanding of the nature of this plant disease and reveals new ways of monitoring and controlling it. The study assessed the effectiveness of preventing treatments by remote sensing, and tracked the pathogenesis of the disease causal agent pathogen, *M. maydis*, in the field plants, using the qPCR molecular detection approach. Specifically, it demonstrated a successful treatment protocol that is based on a sophisticated integration of fungicides mixtures in a schedule adapted to key points in the development of the disease and the use of a fungicide alternation protocol, specifically the replacement of pesticides harboring a different mechanism of action. The remote sensing evaluation supported the results and showed its effectiveness as a research tool for diagnosing infected fields. The pathogen *M. maydis* DNA levels inside the host tissues were in line with the disease symptoms, the plants’ growth parameters, yields (quantity and quality), and the remote thermal sensing measurements.

## Figures and Tables

**Figure 1 jof-06-00054-f001:**
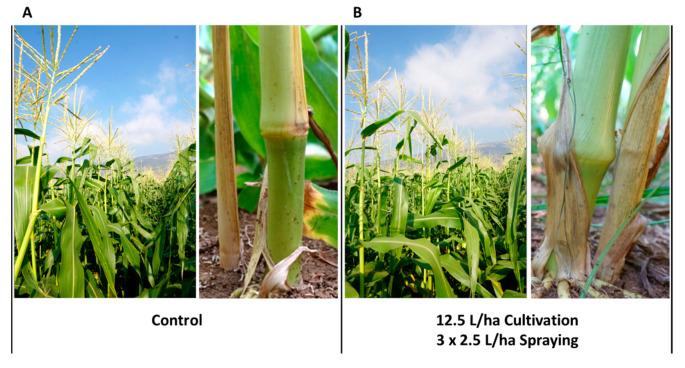
Photograph of the field experiment and stem surface symptoms (Neot Mordechai, spring 2018). Photograph of representative plots of the control (**A**) and the most intensive treatment, 12.5 L/hectare spraying during land tillage, and later three dosages of 2.5 L/hectare (**B**) at the end of the growth season (73 DAS, 18 days after fertilization, DAF) demonstrating that the plants of the entire experimental field had a healthy appearance and typical development. Close inspection of the Jubilee cultivars’ lower stems (near the first above-ground internode) revealed mild symptoms. The stem surface symptoms included partial and occasional color alternation (from green to yellow or brown) of the first above-ground internode and dehydration of the first one or two leaves.

**Figure 2 jof-06-00054-f002:**
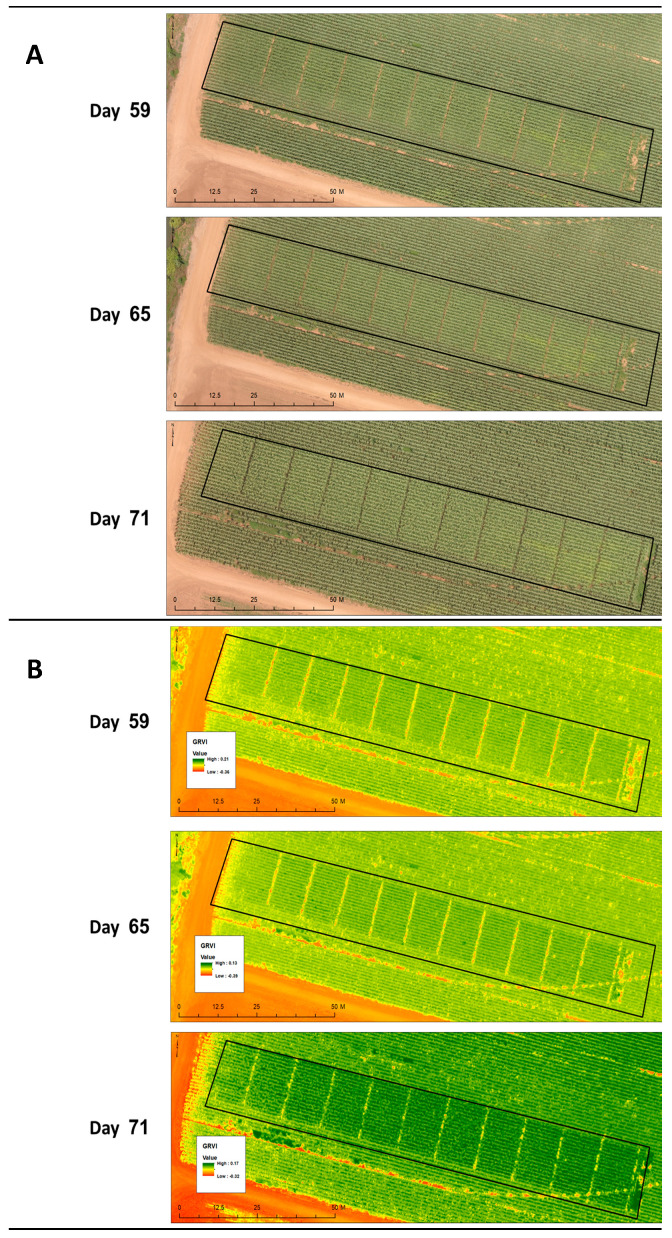
RGB (**A**) and green-red vegetation index (GRVI) (**B**) aerial images of the whole field. The images were taken by a UAV at 59, 65, and 71 DAS (4, 10, and 16 DAF) (Neot Mordechai, spring 2018). Darker green shades in the GRVI image is indicative of a healthy plant while yellow is indicative of a stressed plant, and orange-red indicating soil.

**Figure 3 jof-06-00054-f003:**
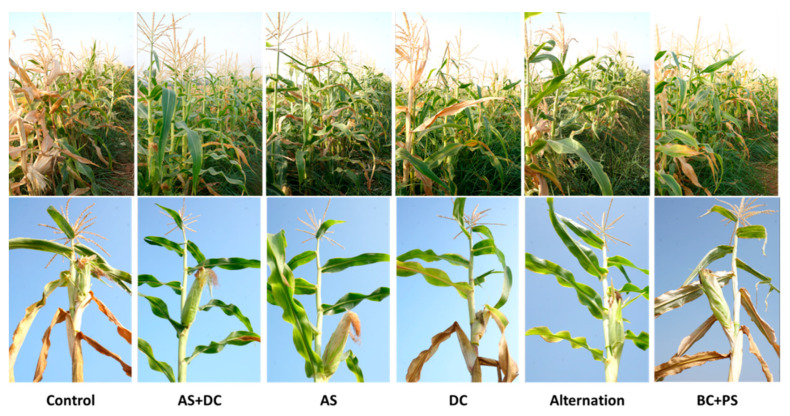
Late wilt disease symptoms in representative plants (Amir, summer 2018). Prelude cultivars’ samples from the field experiment were collected arbitrarily and photographed 71 DAS, 16 DAF. Dehydration symptoms include drying out spreading upwards in the plant, including leaf, stem, and cobs yellowing.

**Figure 4 jof-06-00054-f004:**
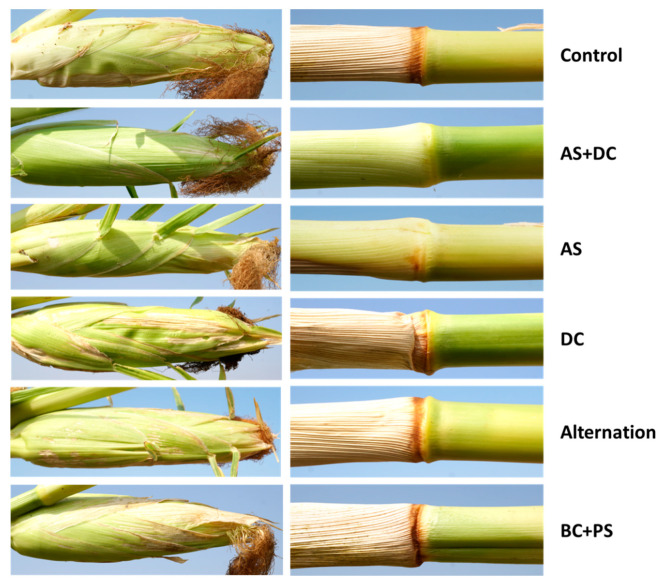
Stem surface and cob symptoms (Amir, summer 2018). Color changes of the plants’ lower stems (near the first above-ground internode, right) and cobs (left). Photos were made of representative plants from the field experiment at 71 DAS (16 DAF).

**Figure 5 jof-06-00054-f005:**
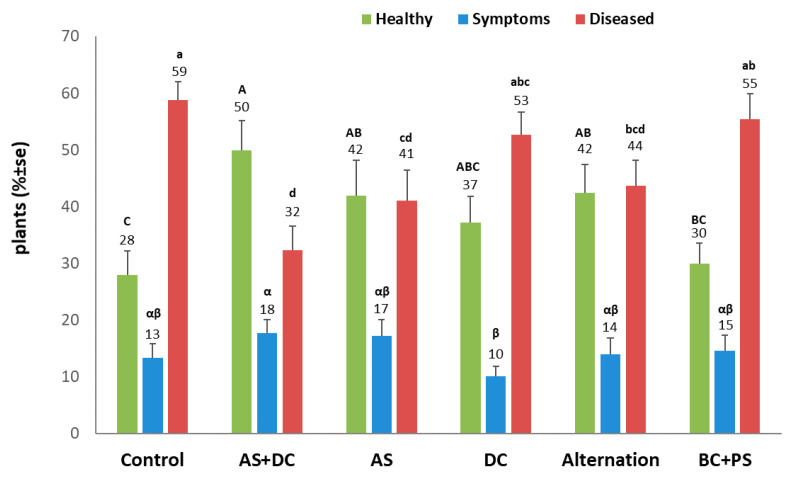
Wilt assessment (Amir, summer 2018). Values were determined 71 DAS (16 DAF). Symptoms – plants with dehydration symptoms appeared on the leaf whose cob was located in its lap. Diseased – plants in which the entire cob dried out. Upper error bars represent the standard error of the mean of 10 replications. Levels not connected by the same letter (A, B, C – for the healthy plants; a, b, c, d – for the diseased plants; and α, β – for the symptomatic plants) are significantly different (*p* < 0.05, ANOVA).

**Figure 6 jof-06-00054-f006:**
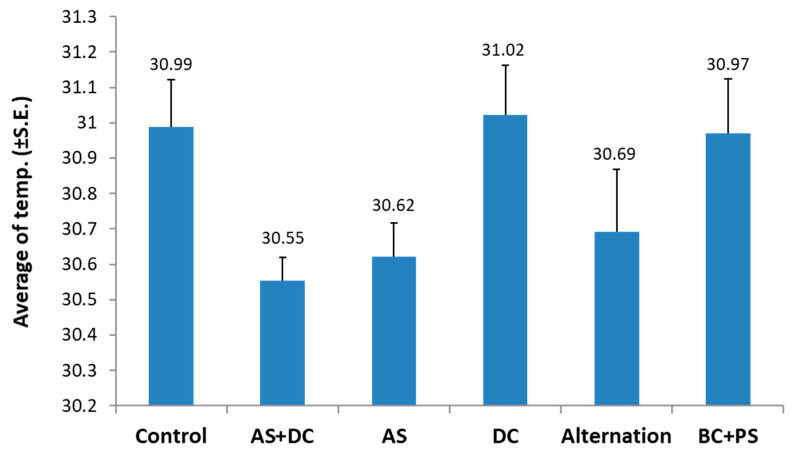
Thermal analysis (Amir, summer 2018). Values indicate an average of 10 repeats. Vertical error bars indicate standard error. No statistical significance between the treatments or between the treatments and the control was found (*p* < 0.05, ANOVA).

**Figure 7 jof-06-00054-f007:**
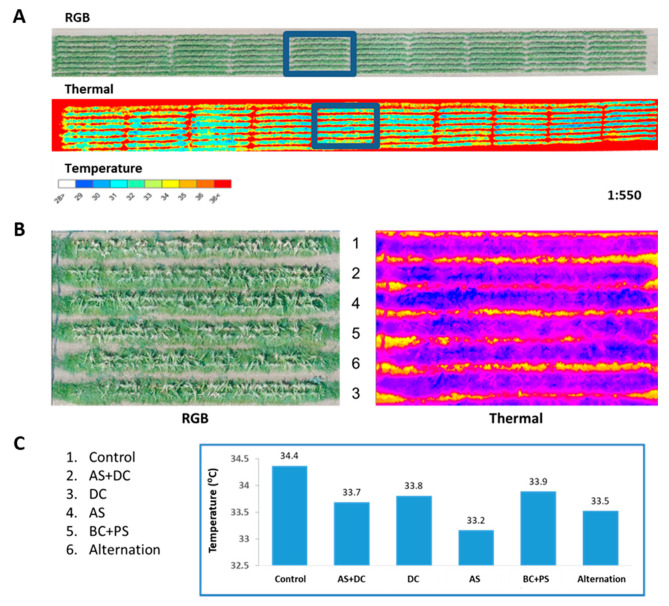
Aerial photograph and thermal analysis sample (Amir, summer 2018). The field was photographed by a quadcopter equipped with an RGB and infra-red sensitive thermal camera 63 DAS (8 DAF). **A**. Aerial overview of the whole field using both cameras, with a scaling ratio of 1:550. The blue rectangles depict block 5. **B**. Close-up of block 5 (from the 10 blocks or repeats) of this experiment. In the RGB aerial photo (left), the AS treatment has a greenish appearance of the canopy with no yellow-brown patches, while the control treatment is seen as yellowish. In the remote aerial imaging of the thermal infra-red crop canopy (right), the temperature difference is observed as a dark-blue (colder temperature) appearance of the AS-treated lines, while the non-protected control lines have a higher temperature represented by a reddish-pink color. **C**. Analyzing block 5 canopy temperature variations among the treatments. Each bar represents one repeat.

**Figure 8 jof-06-00054-f008:**
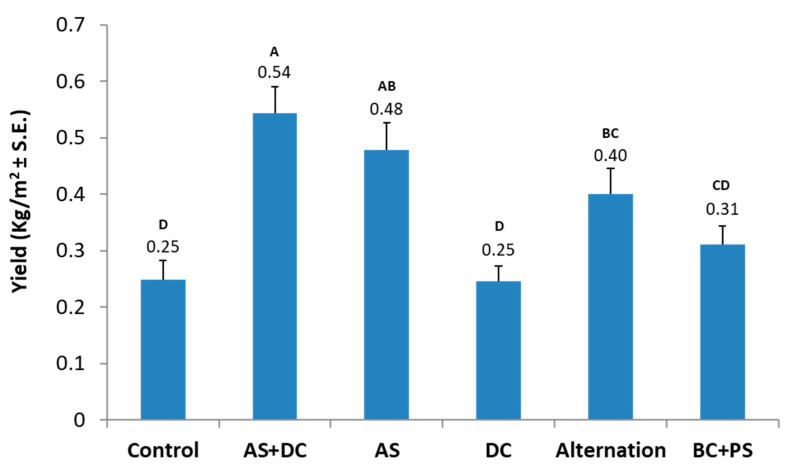
Yield assessment (Amir, summer 2018). The assessment was carried out 21 DAF (76 DAS). Values indicate an average of 10 replications. Upper bars are standard errors. Levels not connected by the same letter (A, B, C, D) are significantly different (*p* < 0.05, ANOVA).

**Figure 9 jof-06-00054-f009:**
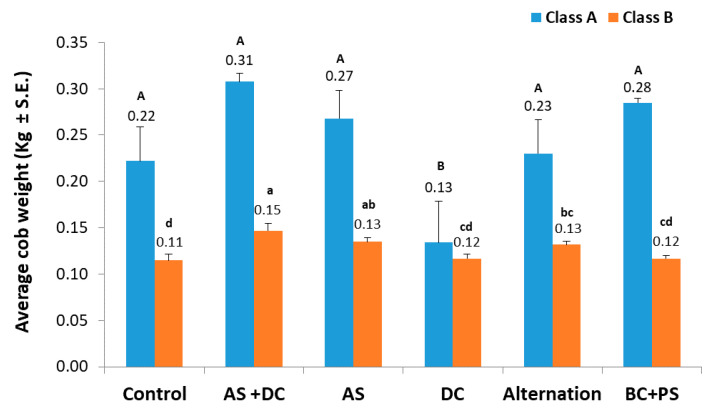
Yield quality (Amir, summer, 2018). The assessment was carried out 21 DAF (76 DAS). The A-class yield classified as cob weight exceeding 250 g. Values represents an average of 10 replications. Upper bars are standard errors. Levels not connected by the same letter (A, B – for the A-class; a, b, c, d – for the B-class) are significantly different (*p* < 0.05, ANOVA).

**Figure 10 jof-06-00054-f010:**
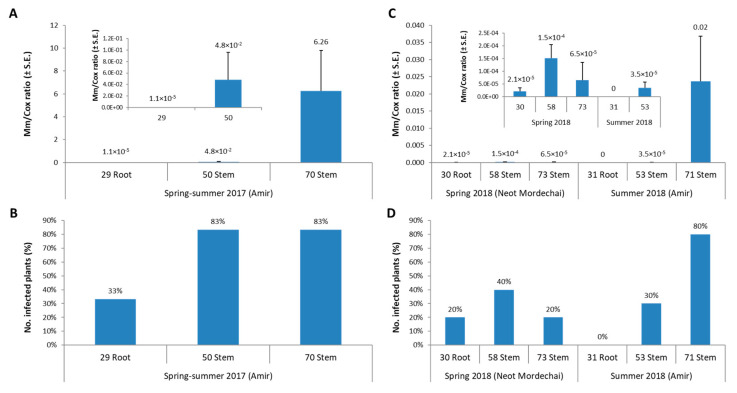
Comparison of qPCR diagnosis of late-wilt pathogenesis. Molecular qPCR tracking of the *M. maydis* DNA at the control treatments in both the spring and summer 2018 field experiments (conducted here, **C**,**D**) compared to the 2017 field experiment results (conducted at the same site) published earlier [37] (**A**,**B**). The y-axis (in **A**,**C**) indicates *H. maydis* proportionate DNA abundance normalized to the cytochrome c oxidase (*COX*) DNA. Bars represent a mean of 10 replicates. Standard errors are indicated in error bars.

**Table 1 jof-06-00054-t001:** Weather conditions in the experimental fields ^a^.

Experiment	Dates	Average Temp.	Min Temp.	Max Temp.	Average Humid.	Min Humid.	Max Humid.	Precipit-Ation
Assessing Azoxystrobin spraying during and after land tillage	23 April–5 July 2018	25 °C	18 °C	36 °C	56%	21%	88%	27 mm
Assessing Azoxystrobin + Difenoconazole seed coating and various fungicides applied by drip irrigation	21 June–5 September 2018	28 °C	21 °C	36 °C	63%	30%	85%	3 mm

^a^ Data from the Kiryat Shmona Academia BaKikar IQIRYATS3 weather station measurements supported by MIGAL–Galilee Research Institute and Tel-Hai College, Israel.

**Table 2 jof-06-00054-t002:** Azoxystrobin treatments in the spraying field experiment (Neot Mordechai, spring 2018) ^a^.

Treatment	Spraying during Land Tillage (L/Hectare)	Spraying 13 DAS ^b^ (L/Hectare)	Spraying 29 DAS (L/Hectare)	Spraying 41 DAS (L/Hectare)
1 – Control ^c^	-	-	-	-
2	5	-	-	-
3	10	-	-	-
4	15	-	-	-
5	20	-	-	-
6	-	2.5	2.5	-
7	5	2.5	2.5	2.5
8	7.5	2.5	2.5	2.5
9	12.5	2.5	2.5	2.5

^a^ Applying the AS fungicide during the land tillage stage was done by spraying followed by mechanical mixing of the top 10–15 cm of the ground before sowing. The chemical spraying was done during the growth session with an electric backpack field sprayer applied to the base of the stem 13, 29 and 41 days after sowing. ^b^ Days after sowing (DAS). ^c^ Control–untreated plots. – no date.

**Table 3 jof-06-00054-t003:** Fungicides used in this study ^a^.

Fungicide Commercial Name and Abbreviations	Manufacturer, Supplier	Active Ingredient (Common Name)	Group Name	Chemical Group	Target Site of Action	Active Ingredient (g/L)	Applied in the Field
**Amistar ** ^**b**^ ** (AS)**	Syngenta (Basel, Switzerland), Adama Makhteshim (Airport City, Israel)	Azoxystrobin (CAS no. 131860-33-8)	QoI-fungicides (quinone outside inhibitors)	Methoxy-acrylates	Respiration C3: cytochrome bc1 (ubiquinol oxidase) at Qo site (*cyt b gene*)	250	Land tillage (5–20 L/hectare) Spraying (2.5 L/hectare × 2/3) Dripline protection (2.25 L/hectare × 3)
**Dividend (DC)**	Syngenta (Basel, Switzerland), Gadot Agro (Kidron, Israel)	Difenoconazole (CAS no. 119446-68-3)	DMI-fungicides (DeMethylation Inhibitors, SBI: Class I)	Triazoles	Sterol Biosynthesis in membranes G1: C14-demethylase in sterol biosynthesis (erg11/cyp51)	30	Dripline protection (2.25 L/hectare × 3)
**Ortiva top ** ^**b**^ ** (AS + DC)**	Syngenta (Basel, Switzerland), Adama Makhteshim (Airport City, Israel)	Azoxystrobin (CAS no. 131860-33-8)	QoI-fungicides (quinone outside inhibitors)	Methoxy-acrylates	Respiration C3: cytochrome bc1 (ubiquinol oxidase) at Qo site (*cyt b gene*)	250	Seed coating 0.002 (mL/seed) Dripline protection (2.25 L/hectare × 3), separately and in fungicide alternation
		Difenoconazole (CAS no. 119446-68-3)	DMI-fungicides (DeMethylation Inhibitors, SBI: Class I)	Triazoles	Sterol Biosynthesis in membranes G1: C14-demethylase in sterol biosynthesis (erg11/cyp51)	125	
**Proline** **+ Folicur ** ^**b**^ **(PR + TE)**	Bayer CropScience (Monheim am Rhein, Germany), Lidorr Chemicals Ltd. (Ramat Hasharon, Israel)	Prothioconazole (Proline) (CAS no. 178928-70-6)	DMI-fungicides (DeMethylation Inhibitors) (SBI: Class I)	Triazolinthiones	Sterol Biosynthesis in membranes G1: C14-demethylation in sterol biosynthesis (*erg11/cyp51*)	275	Dripline protection (2.25 L/hectare) in fungicide alternation
		Tebuconazole (Folicur) (CAS no. 107534-96-3)	DMI-fungicides (DeMethylation Inhibitors) (SBI: Class I)	Triazoles	Sterol Biosynthesis in membranes G1: C14-demethylation in sterol biosynthesis (*erg11/cyp51*)	200	
**Velum + Flint ** ^**b**^ ** (FL + TR)**	Bayer CropScience (Monheim am Rhein, Germany), Lidorr Chemicals Ltd. (Ramat Hasharon, Israel)	Fluopyram (Velum) (CAS no. 658066-35-4)	SDHI (Succinate dehydrogenase inhibitors)	Pyridinyl-ethyl-benzamides	Respiration C2: complex II: succinate-dehydrogenase	200	Dripline protection (2.25 L/hectare) in fungicide alternation
		Trifloxystrobin (Flint) (CAS no. 141517-21-7)	QoI-fungicides (Quinone outside Inhibitors)	Oximino acetates	Respiration C3: complex III: cytochrome bc1 (ubiquinol oxidase) at Qo site (*cyt b gene*)	500	
**Signum ** ^**b**^ ** W.G.** **(BC + PS)**	BASF (Ludwigshafen, Germany), Adama Agan (Ashdod, Israel)	26.7% Boscalid (CAS no. 188425-85-6)	SDHI (Succinate dehydrogenase inhibitors)	Pyridine- carboxamides	Respiration C2: complex II: succinate-dehydrogenase	267	Dripline protection (2.25 L/hectare × 3)
		6.7% Pyraclostrobin (CAS No. 175013-18-0)	QoI-fungicides (Quinone outside Inhibitors)	Methoxy-carbamates	Respiration C3: cytochrome bc1 (ubiquinol oxidase) at Qo site (*cyt b gene*)	67	

^a^ The fungicides information is based on the Fungicide Resistance Action Committee (FRAC) Code List 2018 and the datasheet published by the manufacturer. ^b^ Inspected recently in the field using different application methods [34].

**Table 4 jof-06-00054-t004:** Primers used in this study for *Magnaporthiopsis maydis* detection.

Pairs	Primer	Sequence	Uses	Amplifica-Tion	References
1	A200a-for A200a-rev	5′-CCGACGCCTAAAATACAGGA-3′ 5′-GGGCTTTTTAGGGCCTTTTT-3′	qPCR ^a^	*M. maydis* AFLP ^b^ -derived species-specific fragment	[18]
2	Cox-F Cox-R	5′-GTATGCCACGTCGCATTCCAGA-3′ 5′-CAACTACGGATATATAAGRRCCRRAACTG -3′ ^c^	qPCR control	Cytochrome c oxidase (COX) gene	[40] [42]

^a^ Quantitative real-time PCR (qPCR). ^b^ Amplified fragment length polymorphism (AFLP). ^c^ The R symbol represents Adenine or Guanine (purine). The synthesized primer is comprised of a mixture of primers with both nucleotides.

**Table 5 jof-06-00054-t005:** The efficiency of the Azoxystrobin spraying during and after land tillage (Neot Mordechai, spring 2018) ^a^.

Treatment (L/Hectare)		Emergence (no./m^2^, 17 DAS)		Yield (kg/m^2^, 74 DAS)		Class A (kg/m^2^, 74 DAS)		
		Mean ^c^	S.E.	Mean	S.E.	Mean	S.E.	Percent
1	Control ^b^	10.8 ^AB^	0.58	2.58 ^A^	0.07	2.26 ^A^	0.05	88%
2	5 Tillage	11.4 ^AB^	1.08	2.63 ^A^	0.09	2.39 ^A^	0.08	91%
3	10 Tillage	12.4 ^A^	0.51	2.61 ^A^	0.16	2.19 ^A^	0.13	84%
4	15 Tillage	10.2 ^B^	0.73	2.57 ^A^	0.15	2.32 ^A^	0.10	90%
5	20 Tillage	11.2 ^AB^	0.58	2.57 ^A^	0.18	2.22 ^A^	0.15	86%
6	5 Spraying	10.4 ^B^	0.51	2.32 ^A^	0.09	2.13 ^A^	0.13	92%
7	5 Tillage + 7.5 Spraying	10.4 ^B^	0.75	2.67 ^A^	0.06	2.37 ^A^	0.12	89%
8	7.5 Tillage + 7.5 Spraying	11.8 ^AB^	0.58	2.58 ^A^	0.15	2.15 ^A^	0.09	83%
9	12.5 Tillage + 7.5 Spraying	11.6 ^AB^	0.68	2.73 ^A^	0.19	2.34 ^A^	0.17	86%

^a^ The treatments are detailed in Table 2. The emergence of the seedlings considered to take place when the coleoptile tip appeared above the ground surface. Yield quantity and quality were determined 19 days after fertilization (DAF) and comprised all the upper part plant cobs in each of the experimental plots. The yield classified as class-A yield had a cob weight exceeding 250 g. Values indicate an average of 10 replications. ^b^ Control – untreated plots. ^c^ Levels not connected by the same capital letter (A, B, AB) are significantly different (*p* < 0.05, ANOVA) from the other treatments at the same column.

**Table 6 jof-06-00054-t006:** qPCR diagnosis in the field (Neot Mordechai, spring 2018) ^a^.

Treatment (L/Hectare)	29 DAS (Root)			58 DAS (Stem)			73 DAS (Stem)		
	Mean ^c^	S.E.	Infect. ^d^	Mean	S.E.	Infect.	Mean	S.E.	Infect.
Control ^b^	2.1 × 10^−5 B^	1.4 × 10^−5^	20%	1.5 × 10^−4 B^	5.3 × 10^−5^	40%	6.5 × 10^−5 A^	7.1 × 10^−5^	20%
5 Tillage	1.2 × 10^−3 A^	1.1 × 10^−3^	40%	1.6 × 10^−5 B^	7.1 × 10^−6^	20%	0 ^A^	0	0%
10 Tillage	0 ^B^	0	0%	3.1 × 10^−5 B^	1.3 × 10^−5^	40%	2.8 × 10^−1 A^	2.8 × 10^−1^	60%
15 Tillage	0 ^B^	0	0%	1.6 × 10^−4 B^	5.3 × 10^−5^	40%	9.4 × 10^−4 A^	6.7 × 10^−4^	40%
20 Tillage	0 ^B^	0	0%	7.6 × 10^−5 B^	2.3 × 10^−5^	40%	2.7 × 10^−3 A^	1.1 × 10^−3^	100%
5 Spraying	0 ^B^	0	0%	4.1 × 10^−5 B^	1.8 × 10^−5^	40%	4.6 × 10^−2 A^	4.6 × 10^−2^	40%
5 Tillage + 7.5 Spraying	0 ^B^	0	0%	1.6 × 10^−4 B^	3.3 × 10^−5^	60%	4.1 × 10^−1 A^	4.1 × 10^−1^	20%
7.5 Tillage + 7.5 Spraying	0 ^B^	0	0%	6.0 × 10^−5 B^	2.7 × 10^−5^	20%	2.2 × 10^−3 A^	1.2 × 10^−3^	100%
12.5 Tillage + 7.5 Spraying	0 ^B^	0	0%	4.6 × 10^−4 A^	9.9 × 10^−5^	100%	3.0 × 10^−3 A^	2.0 × 10^−3^	60%

^a^ The treatments are detailed in Table 2. Results are a mean of 10 independent replicates. The fungal DNA levels reflected as *Mm/cox* ratio. ^b^ Control – untreated plots. ^c^ Levels not connected by the same capital letter (A, B) are significantly different (*p* < 0.05, ANOVA) from the other treatments at the same column. ^d^ The infection percentage (prevalence) describes the number of plants (out of 10) that had positive detection using the qPCR method.

**Table 7 jof-06-00054-t007:** The efficiency of the application of the fungicide by dripline irrigation in the field (Amir, summer 2018) ^a^.

Treatment ^c^ (L/Hectare)	Emergence (no./m^2^, 15 DAS)		Root Biomass (mg, 30 DAS)		Shoot Biomass (mg, 30 DAS)	
	Mean	S.E.	Mean	S.E.	Mean	S.E.
Control ^b^	9.3	0.54	4.73	0.70	94.41	12.55
AS + DC	9.6	0.54	4.66	0.78	100.50	11.69
AS	9.0	0.56	5.23	0.81	105.85	10.21
DC	9.1	0.28	6.47	1.01	105.66	9.07
Alternation	9.6	0.40	5.09	1.25	89.94	13.18
BC + PS	8.9	0.41	5.96	1.00	97.46	15.15

^a^ The fungicides are detailed in Table 3. Alternation of fungicides according to the following sequence: (1) AS + DC; (2) Prothioconazole (Proline) and Tebuconazole (Folicur) mixture (PR + TE); and (3) Fluopyram (Velum) and Trifloxystrobin (Flint) mixture (FL + TR). Each fungicide or mixture was applied three times, 18, 31, and 45 DAS at a dosage of 2.25 L/hectare (0.0869 L/treatment). Seedling emergence was considered to occur when the tip of the coleoptile was observed above-ground. The biomass represents wet weight assessment. Results are a mean of 10 independent replicates. ^b^ Control – plants protected only by the AS + DC seed coating. ^c^ No statistical significance between the treatments or between the treatments and the control was found (*p* < 0.05, ANOVA).

**Table 8 jof-06-00054-t008:** qPCR diagnosis in the field (Amir, summer 2018) ^a^.

Treatment ^c^ (L/Hectare)	31 DAS (Root)			58 DAS (Stem)			71 DAS (Stem)		
	Mean	S.E.	Infect.	Mean	S.E.	Infect.	Mean	S.E.	Infect.
Control	0	0	0%	3.5 × 10^−5^	2.2 × 10^−5^	30%	0.024	0.013	80%
AS + DC	1.7 × 10^−4^	1.7 × 10^−4^	20%	2.0 × 10^−3^	2.0 × 10^−3^	20%	0.053	0.042	40%
AS	1.4 × 10^−6^	1.4 × 10^−6^	10%	9.7 × 10^−5^	3.9 × 10^−5^	70%	0.016	0.011	70%
DC	0	0	0%	3.2 × 10^−5^	1.4 × 10^−5^	50%	2.079	2.064	80%
Alternation	0	0	0%	3.3 × 10^−5^	1.3 × 10^−5^	70%	0.001	0.001	60%
BC + PS	2.0 × 10^−4^	1.4 × 10^−4^	40%	9.4 × 10^−4^	8.5 × 10^−4^	70%	0.589	0.780	100%

^a^ The fungicides are detailed in Table 3. The fungal DNA levels reflected as *Mm/cox* ratio. Alternation of fungicides according to the following sequence: (1) AS + DC; (2) Prothioconazole (Proline) and Tebuconazole (Folicur) mixture (PR + TE); and (3) Fluopyram (Velum) and Trifloxystrobin (Flint) mixture (FL + TR). Results are a mean of 10 independent replicates. ^b^ Control – plants protected only by the AS + DC seed coating. ^c^ No statistical significance between the treatments or between the treatments and the control was found (*p* < 0.05, ANOVA).

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
