# Peer review of "Molecular Tracking and Remote Sensing to Evaluate New Chemical Treatments Against the Maize Late Wilt Disease Causal Agent, Magnaporthiopsis maydis"

_jof, 2020, doi:10.3390/jof6020054_

Round 1
Reviewer 1 Report
This manuscripts presents in much details very heavy filed experiments on control of late wilt of maize. Results of fungicide treatments and of disease evaluation by thermal analysis or QPCR of the pathogen DNA may be of interest to the community working on this disease. However, the statistical analyses presented seem insufficient in regard to the huge amount of work deployed, the data set available and the potentially appropriate experimental design used (10 repeats, complete blocks). Correlation analyses are also lacking. Other statistical concerns appear along the result section (see detailed comments). The QPCR evaluation is questionable because the way variables are calculated is unclear and because controls are lacking (or are not presented). There is also a main concern about the occurrence of other diseases in the summer experiment that are not taken into consideration. In particular, their impact on symptoms and on yield are not evaluated. In addition, this article is rather long. It turns out that, in one experiment there was no disease and no conclusions can be drawn. I suggest to remove completely this experiment from the publication.
58 editing problem with the ref. Klaubauf et al.
62 ) is missing.
63 “By flowering” add stage.
66 “…diseased plants had a high…”. Replace had by have
115 replace improve by improved.
139 “These two subsequent experiments…”. Unclear: only one is mentioned in the previous sentence.
164 “Seeds were pretreated with Thiram, Captan, Carboxin, Metalaxyl-M3…”. Unclear: is it a mixture? Which ratio? At least give commercial product name.
224 Name of commercial product? Composition?
264 m or m²? A pixel is supposed to be a surface.
283 “arbitrarily” What do you mean exactly? What was the criterion to choose the plants analyzed?
286 fresh weight?
318 Are primers for COX gene specific of plants?
327 Given the experimental design doing “only” student’s t-test seems inappropriate. At least block effects should be integrated in the statistical model.
332-345 Is this introduction to the result section necessary? It is redundant with the general introduction.
Table 5. Treatment numbers mentioned in Table 2 are not used in Table 5. Making the link is not obvious for some treatments.
395 Percentage increase are not significant and should not be highlighted. The only significant result seems to lower than the control.
Table 6 How was the percentage of infection calculated (% of positive plants by QPCR)? Percentage may not be appropriate if calculated on a limited number (5 or 10). Mean value of which variable? What is the threshold for detection? Where are the negative controls? Which data were used to correlate fungal mycelial quantity and QPCR values (what type of correlation is expected)?
421 Is 10 times significant?
433 Either the difference is significant or it is not. The experimental design and the statistical analyzes are used to take into account the effect of “natural variation”. What is the p value of the test?
Figure 5. Box plot would probably better reflect the real variability of the data. Standard error is minimizing variability (which is expected to be relatively high in field conditions).
Figure 6. Instead of figure 6, a figure showing the correlation between the thermal analysis and the disease evaluation would probably better illustrate the interest of the technique.
493 Why focusing on one block out of ten, ie on the exception rather than on the rule?
532 Are figure numbers correct? Figure 2 is related to the other experiment.
545 What are the correlations between variables? Fungal biomass (estimated by QPCR) vs disease symptom? Disease symptom vs yield? Etc.
574 What was the impact of these other diseases on symptoms and on yield?
Author Response
Responses to Reviewer 1’s comments
We thank the reviewer for investing substantial work that contributes significantly to this manuscript. His/her remarks and suggestions improved this scientific paper remarkably and made it more accurate, clear, focused and well-structured. Your contribution is greatly appreciated.
General comments:
The statistical analyses presented seem insufficient in regard to the huge amount of work deployed, the data set available and the potentially appropriate experimental design used (10 repeats, complete blocks). Correlation analyses are also lacking. Other statistical concerns appear along the result section (see detailed comments).
Indeed, ANOVA with post-hoc such as the Tukey-Kramer HSD test, would be a good option. We chose to do a series of Student’s t-test (which is essentially the same as unprotected Fisher’s LSD test - a set of t-tests, without any correction made for multiple comparisons) because our greatest interest was to discover differences between two groups – each treatment separately compared to the control. To this end, the Student’s t-test can provide a more powerful test (i.e., enabling more easily to accept the test alternative hypothesis – finding differences). Both ways of doing statistics are perfectly good and are customary in the science community.
We believe that this approach is the right choice when interpreting the results of open-field experiments. Such experiments yielded higher variations in the variables than in-door experiments because of changes in the environment and the soil microbiome. This approach was recently used and published in our previous work (see Degani et al., PloS One 2018, 13, e0208353).
The QPCR evaluation is questionable because the way variables are calculated is unclear and because controls are lacking (or are not presented).
The reviewer is correct; this is indeed an important issue that should be clarified.
The specific M. maydis qPCR detection was just recently validated, approved and published (Degani et al., Plant Disease, 2019, 103, 238-248). We have used this qPCR method repeatedly in several additional works (see Degani et al., PloS One 2018, 13, e0208353; Degani et al., Agronomy 2019, 9, 181; Dor and Degani, Plants 2019, 8; Degani et al., Microorganisms 2020, 8). The same protocol with some adjustments is used worldwide in the scientific community for a similar purpose (identifying pathogens’ DNA inside host tissues).
We routinely run positive control in the qPCR reactions that include DNA from a PDA growth culture of M. maydis with bot the cox1 and the M. maydis (A200a) primers and get measurable and relatively constant results.
We calculate the relative gene expression using the mean ΔCt value (threshold cycle). The method we used for calculating relative gene expression from the quantification cycle (Cq) values obtained by the qPCR analysis is explained well in the following citation: Haimes, J., and M. Kelley. “Demonstration of a ΔΔCq calculation method to compute relative gene expression from qPCR data.” Thermo Scientific Tech Note 1 (2010).
The following explanation is present in the text: “The gene encoding the eukaryotic mitochondria respiratory electron transport chain, last enzyme – cytochrome c oxidase (COX) – was used as a reference “housekeeping” gene to normalize the amount of DNA [40]. This gene was amplified using the primer set COX F/R (Table 4). Calculating the relative gene abundance was according to the ΔCt model [41]. The same efficacy was assumed for all samples. All amplifications were performed in triplicate.” (lines 322-326).
Please see below in the specific comments response an explanation about the COX gene used as “housekeeping” and reference gene.
There is also a main concern about the occurrence of other diseases in the summer experiment that are not taken into consideration. In particular, their impact on symptoms and on yield are not evaluated.
Indeed, the soil most certainly contains other microorganisms, some of which may be pathogenic fungi. However, the non-treated control group was used as a reference group, and all experimental plots were compared to this control. Thus, if any other factors, such as other pathogens and environmental conditions, are reflected in the plant’s health, development and yield, they are common to the control and all the treatments. The measurements made prove that the unique disease symptoms outcome and the molecular specific targeting of M. maydis pathogens’ DNA spreading inside the host tissues are the consequences of the treatments.
Still, the possible influence of other microorganisms in the soil in this regard is very interesting. It should be the focus of follow-up work that will examine this question thoroughly. Our group recently demonstrated such interactions (see Degani et al., Microorganisms 2020, 8).
The following sentence was added to the Introduction: “Stalk symptoms may be worsened by secondary invaders such as Fusarium verticillioides causing stalk rot, and Macrophomina phaseolina causing charcoal rot [9], although an antagonistic relationship may exist [6,10].” (lines 51-54)
Also, the following paragraph in the Discussion was edited to elaborate this: “As mentioned in the description of the results, maize crop yield of over 2 kg/m2 is considered normal in healthy fields. The spring 2018 experiment (Neot Mordechai field) achieved high yields of 2.5 kg/m2, but in the subsequent 2018 summer experiment (Amir field), the untreated controls resulted in poor yields of 0.25 kg/m2 (Fig 8), and even the most effective treatment, AS+DC drip protection, recovered cob yield to only 0.54 kg/m2. Despite this significant improvement in yield value, this harvest production is still considered relatively low in commercial fields. Since there were no drastic changes in weather conditions (Table 1), this difference between the spring and summer control treatments was probably the result of other factors. These factors are probably are results of the environment since both maize cultivars studied here (the Jubilee cv. and the Prelude cv.) can reach similar yield production (kg/m2), as presented in this work and previous work [37]. Indeed in the summer experiment, the field suffered from another fungal disease caused by Fusarium verticillioides and Fusarium oxysporum, which led to dehydration and yield loss. Nevertheless, in the plants that were not affected by the Fusarium spp. disease, the drip protection with AS+DC abolished almost completely any sign of late wilt disease. This effect is probably not reflected in the results' interpretation since the non-treated control group was used as a reference group, and all experimental plots were compared to this control. Thus, if any other factors, such as other pathogens and environmental conditions, are reflected in the planet’s health, development and yield, they are common to the control and all the treatments, and equally reflected in the results.” (lines 581-598)
In addition, this article is rather long. It turns out that, in one experiment there was no disease and no conclusions can be drawn. I suggest to remove completely this experiment from the publication.
Indeed, it is possible to shorten the manuscript, as suggested by the reviewer. Nevertheless, the second reviewer and the third reviewer did not recommend this, and we agree with them. We believe that the spring experiment provides essential and supporting information about the disease and M. maydis pathogenesis, and serves as an example of an apparently healthy field. It is a unique situation that allows us to compare a healthy field to a diseased field, while implementing preventive treatments. The two fields were studied thoroughly using the molecular qPCR-based method and remote sensing. Their combined results are advancing our understanding of the pathogen spread inside the host tissues in those two extreme situations. Thus, we think it is important to present both experiments to deepen our understanding of the pathogen colonization ability during pathogenesis in various field disease burst situations.
We elaborate this in the Discussion: “Tracking the M. maydis DNA fluctuations using the qPCR method proven earlier [3] exhibited interesting results. By examining only the control treatment in both field experiments conducted here (spring and summer 2018), a different pattern is revealed (Fig. 10). In both experiments, the pathogen DNA in the stem at 53-58 DAS was higher than the levels in the roots at 30-31 DAS. However, in the healthy state Neot Mordechai field plants (spring 2018), this trend changed, and M. maydis DNA levels dropped at the session end (73 DAS). In contrast, in the heavily diseased Amir field in the summer of the same year, the pathogen’s DNA levels intensified towards the session end (71 DAS). It is logical to assume that the decrease in DNA levels in the non-diseased plants and the sharp increase in these levels in the diseased plants are related to plant health and its immune system ability to prevent fungal spread. Indeed, it was shown that the susceptibility of sensitive maize cv. to late wilt decreases with age [7]. Thus, recovery from late wilt in slightly or moderately infested areas may be associated with a decrease in fungal DNA levels inside the host tissue. In contrast, the weakening of the plants in severe cases of late wilt may lead to the opposite tendency, with sharp fungal DNA levels elevation within the host. To support this, comparing the summer field experiment 2018 results to the summer 2017 results (conducted at the same site) published earlier [37] indicates that in both cases, the harsh disease outbreak was accompanied by a sharp elevation in fungal DNA inside the stem tissue (Fig. 10).” (lines 599-615).
Specific comments:
58 editing problem with the ref. Klaubauf et al.
The sentence was corrected, as advised. The enire paragraph was edited and now reads: “The genus Magnaporthiopsis was established by Luo and Zhang [13] to group three pathogenic species, M. poae, M. incrustans and M. rhizophila, which had previously belonged to the genera Magnaporthe and Gaeumannomyces. M. maydis was recently transferred to this genus from Harpophora by Klaubauf et al. [14].” (lines 57-60).
62 ) is missing.
Corrected, as advised.
63 “By flowering” add stage.
The missing information was added: “By flowering (R1-silking, silks visable outside the husks), approximately 50-60 days from sowing, the first symptoms were gradually revealed.” (lines 65-66).
66 “…diseased plants had a high…”. Replace had by have
Corrected, as advised.
115 replace improve by improved.
Corrected, as advised.
139 “These two subsequent experiments…”. Unclear: only one is mentioned in the previous sentence.
The sentence was corrected as advised to: “Two subsequent field experiments for assessing fungicide efficiency in controlling M. maydis pathogenesis in susceptible cultivars of sweet corn were conducted one after the other during the spring and summer of 2018.” (lines 139-141)
164 “Seeds were pretreated with Thiram, Captan, Carboxin, Metalaxyl-M3…”. Unclear: is it a mixture? Which ratio? At least give commercial product name.
This information was already stated in detailes in the text: “Seeds pretreated with Thiram, Captan, Carboxin, Metalaxyl-M (manufactured by Rogers/Syngenta Seeds, Boise, ID, USA, were supplied by CTS, Tel Aviv, Israel, quota NC7323XLF)” (lines 165-167). The “Thiram, Captan, Carboxin” is the common term used for this seed protection treatment.
224 Name of commercial product? Composition?
The missing information was added to the text: “Seeds pretreatment was done according to a standard commercial procedure by Gadot Agro, Kidron, Israel, with a 0.002 cm3/seed AS+DC commercial preparation (Ortiva top, see Table 3), as recommended by the manufacturer – Syngenta AG, Basel, Switzerland.” (lines 224-226)
264 m or m²? A pixel is supposed to be a surface.
Corrected to M2 as suggested.
283 “arbitrarily” What do you mean exactly? What was the criterion to choose the plants analyzed?
The following explanation was added to the text to clarify this point: “The plants were collected from scattered places along the row, and special care was taken to ensure they had the same phenological development and disease characteristics as their nearby plants (thus, they well represent the average plants in their surroundings).” (lines 286-288)
286 fresh weight?
Indeed, fresh weight. Corrected as advised.
318 Are primers for COX gene specific of plants?
Cytochrome c oxidase catalyzes the transfer of electrons from cytochrome c to oxygen during the final step of the respiratory chain (reviewed by Capaldi et al. 1983). The protein is ubiquitous to all aerobic cells. In eukaryotes, the enzyme is localized to the mitochondrial inner membrane. Fungal mitochondrial genomes usually harbor 14 core-genes encoding proteins involved in electron transport and oxidative phosphorylation, including the cytochrome c oxidase (cox1, cox2, cox3).
See, for example, reference:
Franco MEE, López SMY, Medina R, Lucentini CG, Troncozo MI, et al. (2017) The mitochondrial genome of the plant-pathogenic fungus Stemphylium lycopersici uncovers a dynamic structure due to repetitive and mobile elements. PLOS ONE 12(10): e0185545. https://doi.org/10.1371/journal.pone.0185545
327 Given the experimental design doing “only” student’s t-test seems inappropriate. At least block effects should be integrated in the statistical model.
The statistical method chosen and the rationale for this approach are explained in detail in our response to the general comments above.
332-345 Is this introduction to the result section necessary? It is redundant with the general introduction.
We were asked by reviewer 3 to point out the specificity of this study and explain its continuation to the existing scientific knowledge. However, in light of the above suggestion, we rephrased and drastically shortened the result opening paragraph.
Table 5. Treatment numbers mentioned in Table 2 are not used in Table 5. Making the link is not obvious for some treatments.
Treatment numbers were added to Table 5, as suggested.
395 Percentage increase is not significant and should not be highlighted. The only significant result seems to lower than the control.
The reviewer is correct. The sentence was rephrased to describe the results more accurately: “A statistically insignificant addition of 6% of cob yield resulted in the highest levels of chemical treatment applied. This improvement was also reflected by a 4% increase (not statistically significant) in yield quality (number of A-class cobs that weighed more than 250 g).” (lines 399-401)
Table 6 How was the percentage of infection calculated (% of positive plants by QPCR)? Percentage may not be appropriate if calculated on a limited number (5 or 10). Mean value of which variable? What is the threshold for detection? Where are the negative controls? Which data were used to correlate fungal mycelial quantity and QPCR values (what type of correlation is expected)?
Field pathogenicity trials that rely on natural soil infestation have highly variable results. Even in heavily infested fields, the spreading of the pathogen is not uniform. The pathogen is scattered in small quantities in the soil and the disease spreading in the field is not uniform (see, for example, Degani et al., Agronomy 2019, 9, 181, Figure 1). Moreover, The variables are subject to changes in environmental conditions.
The qPCR method we used is very sensitive and capable of detecting variations in the amount of the pathogens’ DNA inside the host plant tissues with a million-fold difference (see, for example, Degani et al., PloS One 2018, 13, e0208353). Nevertheless, in fields that have moderate or minor disease outbursts (as in the Neot Mordechai spring 2018 experiment), some of the measurements resulted in zero values.
Since the mean of the results is affected by extreme values, we use two measures to describe the results: severity (mean value) and prevalence (percentages of infected plants). This is crucial for an accurate description of the results.
In this case, the infection percentage (prevalence) describes the number of plants (out of 10) that had positive detection using our qPCR method. This percentage supports the DNA relative values (severity) and provides a better understanding of the results.
The following explanation was added to Table 6 footnotes: “The infection percentage (prevalence) describes the number of plants (out of 10) that had positive detection using the qPCR method.”
The qPCR results calculation, controls, and interpretation are explained in our response to the general comments above.
When quantifying the pathogen DNA using the Real-Time PCR assay, as was done in this study, it is important to note that the correlation between DNA quantity and symptom development has only now started to be revealed. To date, there were several cases when the PCR molecular method detected the presence of the pathogen in the host tissues, regardless of treatment success (see Degani et al., Phytoparasitica 2014; 42). Introducing the sensitive and accurate qPCR method is now enabling a better understanding of these connections. Here, the relationship between the observed levels of pathogen DNA in different plant tissues and plant development stages to ultimate disease incidence was demonstrated and discussed (see lines 599-634).
421 Is 10 times significant?
As detailed above in our response to the former question, in the field trials’ molecular DNA tracking, a high level of variations exists within the results due to changes in environmental conditions and the non-uniform spreading nature of the late wilt disease pathogen (Degani et al., Agronomy 2019, 9, 181). Consequently, relatively high standard error values resulted, and in most of those tests, no statistically significant difference could be measured in comparison to the control. This explanation was added to the text (lines 336-340).
433 Either the difference is significant or it is not. The experimental design and the statistical analyzes are used to take into account the effect of “natural variation”. What is the p value of the test?
The sentence was modified and now stated “A statistically significant difference from the control was identified only in the fungicide alternation treatment (P < 0.05)…” (lines 440-442)
Figure 5. Box plot would probably better reflect the real variability of the data. Standard error is minimizing variability (which is expected to be relatively high in field conditions).
Indeed, both ways of presenting the data are common and correct. We prefer to leave the figure as is at the moment. The figure format can be easily altered at a later stage according to the editor's decision.
Figure 6. Instead of figure 6, a figure showing the correlation between the thermal analysis and the disease evaluation would probably better illustrate the interest of the technique.
We believe Figure 6 serves precisely for this purpose. If you compare Figure 6 to Figure 5 (Disease plants pattern), you can see an approximate mirror image. Figure 5 results also very nicely fit the yield assessment results presented in Figure 8.
493 Why focusing on one block out of ten, ie on the exception rather than on the rule?
Block 10 exemplifies very nicely the overall results presented in Figure 6 and serves as a thermal analysis sample. We think this example is sufficient to demonstrate the analysis technique we used here.
532 Are figure numbers correct? Figure 2 is related to the other experiment.
Indeed, the figure numbers are correct – Figures 1 and 2 present the results of the Neot Mordechai spring 2018 experiment, while Figures 3-9 present the results of the Amir summer 2018 experiment.
545 What are the correlations between variables? Fungal biomass (estimated by QPCR) vs disease symptom? Disease symptom vs yield? Etc.
The paragraph was edited and rephrased, as suggested. It now reads: “At the end of the growth session (71 DAS), all the chemical protective treatments that contained AS (especially AS-DC) reduced the number of infected plants according to the qPCR evaluation. The molecular data are in agreement with the disease symptoms recording (Fig. 5) and yield results (Figs. 8-9). The achievement of the AS-DC treatment in limiting the disease symptoms’ severity (Figs. 2-5) was reflected in the pathogen DNA spread from day 53 in the stem onwards (Table 8). DNA extracted from the plant samples collected at 53 and 71 days from sowing for subsequent analysis using the qPCR-based technique revealed that the AS+DC treatment reduced the infected plant's percentage by 33% and 50%, respectively.” (lines 547-554)
574 What was the impact of these other diseases on symptoms and on yield?
The paragraph was rephrased to clarify this point: “Despite this significant improvement in yield value, this harvest production is still considered relatively low in commercial fields. Since there were no drastic changes in weather conditions (Table 1), this difference between the spring and summer control treatments was probably the result of other factors. These factors are probably are results of the environment since both maize cultivars studied here (the Jubilee cv. and the Prelude cv.) can reach similar yield production (kg/m2), as presented in this work and previous work [37]. Indeed in the summer experiment, the field suffered from another fungal disease caused by Fusarium verticillioides and Fusarium oxysporum, which led to dehydration and yield loss. Nevertheless, in the plants that were not affected by the Fusarium spp. disease, the drip protection with AS+DC abolished almost completely any sign of late wilt disease.” (lines 58-594)

Reviewer 2 Report
The manuscript is very interesting, focusing one of the majors contains to maize production in several countries, giving an important contribute in the search for maize late wilt disease management practices, that include chemical treatments and cultural management. It is written in a very clearly way. The approach used is very interesting and the methods are appropriate.
My major concern regards the molecular diagnosis of the late-wilt pathogen, through qPCR.
I understand that the A200 primer set was designed on an AFLP fragment that was previously proved to be species-specific. However, my concern is if it was ever performed by the authors or in the related publications refereed by the authors (Degani et al. 2019, Drori et al. 2013), cross-reactivity tests. This means that, in order to confirm the specificity of the method, it should be demonstrated that the primers should only amplify DNA of the target species, in this case the M. maydis, and not other fungi that might be present in maize, in order to avoid the occurrence of false-positives. The analysis of melting curve it is not, in my opinion, so straight forward. It might happen, that although the AFLP fragment is specific for M. maydis, the primers were designed in locations not so specific. I performed a quick bioinformatic search and it appears that A200 primers have high homology with other Fusarium species. Even more, qPCR was performed through a SyberGreen technology instead of the high specific TaqMan probes chemistry.
I would like to have the authors comments regarding this issue.
Specific comments:
- In the abstract: you refer to the ‘first experiment’ but not to the ‘second experiment’.
- Line 142-142: I understand that the first experiment evaluated the Azoxystrobin spraying in spring, and the second experiment evaluated Azoxystrobin + Difenoconazole seed coating and various fungicides in summer. It seems by your sentence that each experiment evaluates both conditions. Please rephrase the sentence.
- The title ‘2.4. DNA extraction and qPCR’, should be considered as a sub-title of ‘2.3. Molecular diagnosis of the late-wilt pathogen’.
My suggestion is:
2.3. Molecular diagnosis of the late-wilt pathogen
2.3.1. Plant material
2.3.2. DNA extraction and qPCR
2.3.2.1. DNA extraction
2.3.2.2. qPCR-based method
Author Response
Responses to Reviewer 2’s comments
We would like to express our sincere appreciation to the reviewer for important and helpful suggestions and advice. The time and effort invested are greatly appreciated, and without a doubt, contributed to the manuscript and significantly improved it. Thank you.
General comments:
My major concern regards the molecular diagnosis of the late-wilt pathogen, through qPCR.
I understand that the A200 primer set was designed on an AFLP fragment that was previously proved to be species-specific. However, my concern is if it was ever performed by the authors or in the related publications refereed by the authors (Degani et al. 2019, Drori et al. 2013), cross-reactivity tests. This means that, in order to confirm the specificity of the method, it should be demonstrated that the primers should only amplify DNA of the target species, in this case the M. maydis, and not other fungi that might be present in maize, in order to avoid the occurrence of false-positives. The analysis of melting curve it is not, in my opinion, so straight forward. It might happen, that although the AFLP fragment is specific for M. maydis, the primers were designed in locations not so specific. I performed a quick bioinformatic search and it appears that A200 primers have high homology with other Fusarium species. Even more, qPCR was performed through a SyberGreen technology instead of the high specific TaqMan probes chemistry.
I would like to have the authors' comments regarding this issue.
We agree with the reviewer that this concern should be addressed. First, the A200 resultant amplified sequence is a major part of a larger AFLP fragment that had already be proven earlier to be species-specific by Saleh et al. (2003) [1] and Zeller et al. (2000) [2]. The specific M. maydis detection was reported by us for PCR in 2013 (Drori et al., Phytopathologia Mediterranea, 2013) and was just recently validated for qPCR, approved and published (Degani et al., Plant Disease, 2019). Moreover, the specific molecular method was tested and used by us in seven additional publications [3] in leading scientific journals.
These species-specific primers were tested and validated in the past decade in our lab in in vitro experiments with a single pathogen inoculation (pure cultures, seed assay, detached root assay) in seedlings in a growth chamber and in full growth season experiments in a greenhouse. They were also used in commercial fields with naturally infested soils, as presented in this work. All these experiments were conducted under strict experimental design and were well controlled. Negative uninfected controls repeatedly resulted in axenic tissues and zero levels of detection with the molecular method. Moreover, the levels of M. maydis DNA measured throughout the season were in correlation with the maize late wilt disease-specific symptom severity (including dehydration symptoms and vascular tissue occlusion) and outcome (their effect on plant phenological development, yield production and yield quality).
In addition, we tested the ability of the A200 primers to detect Fusarium sp. (probably F. verticillioides) isolated from commercial field disease maize plants [4]. The molecular PCR assay was clearly able to identify M. maydis, but not the Fusarium isolate.
Details of corrections:
In the abstract: you refer to the ‘first experiment’ but not to the ‘second experiment’.
Corrected as advised (lines 24-27): “In the second experiment conducted in the following summer of the same year in a nearby field, the disease outbreak was dramatically higher, with about 350 times higher levels of the pathogen DNA in the untreated plots’ plants.”
Line 142-142: I understand that the first experiment evaluated the Azoxystrobin spraying in spring, and the second experiment evaluated Azoxystrobin + Difenoconazole seed coating and various fungicides in summer. It seems by your sentence that each experiment evaluates both conditions. Please rephrase the sentence.
The sentence e was rephrased to clarify this issue: “Two subsequent field experiments for assessing fungicide efficiency in controlling M. maydis pathogenesis in susceptible cultivars of sweet corn were conducted one after the other during the spring and summer of 2018.” (lines 139-141)
The title ‘2.4. DNA extraction and qPCR’, should be considered as a sub-title of ‘2.3. Molecular diagnosis of the late-wilt pathogen’.
My suggestion is:
2.3. Molecular diagnosis of the late-wilt pathogen
2.3.1. Plant material
2.3.2. DNA extraction and qPCR
2.3.2.1. DNA extraction
2.3.2.2. qPCR-based method
Corrected as advised.
[1] Saleh, A. A. et al. Amplified fragment length polymorphism diversity in Cephalosporium maydis from Egypt. Phytopathology 93, 853-859 (2003).
[2] Zeller, K. A., Jurgenson, J. E., El-Assiuty, E. M. & Leslie, J. F. Isozyme and amplified fragment length polymorphisms from Cephalosporium maydis in Egypt Phytoparasitica 28, 121-130 (2000).
[3] Degani, O. & Cernica, G., Advances in Microbiology (2014), Degani et al., Phytoparasitica (2014), Degani et al., Physiology and Molecular Biology of Plants (2015), Degani et al. PloS One (2018), Degani et al., Agronomy (2019), Dor, S. & Degani, Plants (2019), Degani et al., Microorganisms (2020).
[4] Drori, R. Involvement of Harpophora maydis in late wilt disease of sweet corn: Characterization of the disease cycle and identifying means of control. Master in Science of Agriculture thesis, Hebrew University of Jerusalem (2010).

Reviewer 3 Report
The subject of the manuscript is consistent with the scope of the Journal. The present paper is prepared in the usual manner for scientific work, both the division into chapters collected results in the form of tables. The authors applied correct analytical methods and received many interesting results. The obtained results usually do not raise any substantive or scientific objections. The results are correctly interpreted and developed.
I agree with the authors that research of this type is important. The authors should make an effort to point out specificity of this study, and what is its continuation to the scientific existing knowledge.
Please better explain why these studies were taken, what new they would bring to science.
Please, be sure that all the references cited in the manuscript are also included in the reference list and vice versa with matching spellings and dates.
Author Response
Responses to Reviewer 3’s comments
We thank the reviewer for investing time and effort, which contributed to this manuscript. The helpful and necessary remarks and suggestions improved this scientific paper and made it more accurate, clear and focused.
General comments:
The authors should make an effort to point out the specificity of this study, and what is its continuation to the scientific existing knowledge. Please better explain why these studies were taken, what new they would bring to science.
This is indeed an important aspect that should be stated clearly. The novelty of the research and its contribution to the scientific knowledge in the subject matter was clarified in several places throughout the text:
- Abstract (lines 17-18): “The current study aimed at advancing our understanding of the nature of this plant disease and revealing new ways to monitor and control it.”
- Introduction (lines 112-115): “The current work follows the 2009-2010 [34] and 2017 [37] It aims at addressing three main topics: i) improve preventive treatments to cope with the disease; ii) study M. maydis pathogenesis by tracking the pathogen DNA in the plants' inner-tissues and plants' outer symptoms; and iii) remote sensing to monitor the plants' health and evaluate the effectiveness of the treatments.”
- Materials and Methods (lines 143-146): “Both experiments aimed at evaluating Azoxystrobin as a sole fungicide or in combination with Difenoconazole, examining new ways of applying the preparation, assessing its effectiveness in comparison to other pesticides, and developing a method that would prevent the development of resistance against the preparation.”
- Results section opening paragraph (lines 342-350): “By using molecular targeting of the pathogen DNA inside the plants' tissues, above-ground symptoms evaluation and new remote sensing of the plants' health, we conducted two large field experiments to inspect new chemical treatments against the maize late-wilt pathogen, maydis. The preventive treatment includes applying the fungicide during and after land tillage by spraying it on the base of the stem at a timetable fitting the pathogen lifecycle key points that succeeded earlier in restricting the disease outburst. This application is important to the growth area in which watering is done using a frontal irrigation system. The current work also tested the application of different anti-fungal formulations through irrigation driplines and the alternation of pesticides in order to prevent the development of fungicide resistance.”
- Conclusions section (lines 667-680).
Please, be sure that all the references cited in the manuscript are also included in the reference list and vice versa with matching spellings and dates.
The reviewer is correct; this issue should be double-checked. We made sure that all the references cited in the manuscript are included in the reference list and vice versa.

Round 2
Reviewer 1 Report
The authors have done minor text editing. But they did not consider doing major revision of the manuscript as suggested. Answers to comments are not convincing.
Author Response
Responses to Reviewer 1’s report 2 comments
We would like to express our appreciation to you for the important and helpful corrections, suggestions, and advice. This contribution significantly improved the manuscript. Thank you.
The statistical analyses presented seem insufficient in regard to the huge amount of work deployed, the data set available and the potentially appropriate experimental design used (10 repeats, complete blocks). Correlation analyses are also lacking. Other statistical concerns appear along the result section (see detailed comments).
As recommended by you and the Academic Editor, we have redone and replaced all the statistical analyses presented in the manuscript.
The one-way ANOVA followed by multiple comparisons posthoc of the student’s t-test for each pair (with correction for multiple comparisons) was used to evaluate the M. maydis infection outcome in the experiments and the effectiveness of the treatments.
Data analysis and statistics were done using the JMP program, 15th edition, SAS Institute Inc., Cary, NC, USA.
The following changes were made to the manuscript following this change:
- Section 2.5 – ‘Statistical analyses’ was rewritten.
- Tables 5, 6, 7, 8, and their footnotes were updated.
- Figures 5, 6, 8, 9, and their legends were updated.
- The results statistic description (throughout the Result section) was updated.
In addition, this article is rather long. It turns out that, in one experiment there was no disease, and no conclusions can be drawn. I suggest to remove completely this experiment from the publication.
Since this issue is controversial among the reviewers (you recommended to remove the spring experiment entirely from the manuscript, while the other two reviewers didn’t recommend this), we will leave the decision to the editor.
We elaborated on this in our previous response letter:
Indeed, it is possible to shorten the manuscript, as suggested by the reviewer. Nevertheless, the second reviewer and the third reviewer did not recommend this, and we agree with them. We believe that the spring experiment provides essential and supporting information about the disease and M. maydis pathogenesis, and serves as an example of an apparently healthy field. It is a unique situation that allows us to compare a healthy field to a diseased field, while implementing preventive treatments. The two fields were studied thoroughly using the molecular qPCR-based method and remote sensing. Their combined results are advancing our understanding of the pathogen spread inside the host tissues in those two extreme situations. Thus, we think it is important to present both experiments to deepen our understanding of the pathogen colonization ability during pathogenesis in various field disease burst situations.
We elaborate this in the Discussion: “Tracking the M. maydis DNA fluctuations using the qPCR method proven earlier [3] exhibited interesting results. By examining only the control treatment in both field experiments conducted here (spring and summer 2018), a different pattern is revealed (Fig. 10). In both experiments, the pathogen DNA in the stem at 53-58 DAS was higher than the levels in the roots at 30-31 DAS. However, in the healthy state Neot Mordechai field plants (spring 2018), this trend changed, and M. maydis DNA levels dropped at the session end (73 DAS). In contrast, in the heavily diseased Amir field in the summer of the same year, the pathogen’s DNA levels intensified towards the session end (71 DAS). It is logical to assume that the decrease in DNA levels in the non-diseased plants and the sharp increase in these levels in the diseased plants are related to plant health and its immune system ability to prevent fungal spread. Indeed, it was shown that the susceptibility of sensitive maize cv. to late wilt decreases with age [7]. Thus, recovery from late wilt in slightly or moderately infested areas may be associated with a decrease in fungal DNA levels inside the host tissue. In contrast, the weakening of the plants in severe cases of late wilt may lead to the opposite tendency, with sharp fungal DNA levels elevation within the host. To support this, comparing the summer field experiment 2018 results to the summer 2017 results (conducted at the same site) published earlier [37] indicates that in both cases, the harsh disease outbreak was accompanied by a sharp elevation in fungal DNA inside the stem tissue (Fig. 10).” (lines 599-615).

Reviewer 2 Report
The authors clarifed the points that were raised. According to me the manuscript can now be published.
Author Response
We would like to express our appreciation to you for the important and helpful corrections, suggestions, and advice. This contribution significantly improved the manuscript. Thank you.
